# Spike Accumulation Forwarding for Effective Training of Spiking Neural Networks

## Abstract

In this article, we propose a new paradigm for training spiking neural networks (SNNs), *spike accumulation forwarding* (*SAF*). It is known that SNNs are energy-efficient but difficult to train. Consequently, many researchers have proposed various methods to solve this problem, among which online training through time (OTTT) is a method that allows inferring at each time step while suppressing the memory cost. However, to compute efficiently on GPUs, OTTT requires operations with spike trains and weighted summation of spike trains during forwarding. In addition, OTTT has shown a relationship with the Spike Representation, an alternative training method, though theoretical agreement with Spike Representation has yet to be proven. Our proposed method can solve these problems; namely, SAF can halve the number of operations during the forward process, and it can be theoretically proven that SAF is consistent with the Spike Representation and OTTT, respectively. Furthermore, we confirmed the above contents through experiments and showed that it is possible to reduce memory and training time while maintaining accuracy.

## 1 Introduction

Due to the carbon emission reduction problem, energy-efficient spiking neural networks (SNNs) are attracting attention (Luo et al., 2023). SNNs are known to be more bio-plausible models than artificial neural networks (ANNs) and can replace the multiply-accumulate (MAC) operations with additive operations. This characteristic comes from propagating the spike train (belonging to $\{0,1\}^T$, where $T$ is the number of time steps) and is energy-efficient on neuromorphic chips (Akopyan et al., 2015; Davies et al., 2018).

Despite the usefulness of SNNs for $CO_2$ reduction, their neurons are non-differentiable, which makes them difficult to train. Solving this problem is in the mainstream of SNN research, and back-propagation through time (BPTT) with surrogate gradient (SG) (Zheng et al., 2021; Xiao et al., 2022) is one of the main methods to achieve high performance. In particular, the recently proposed Online Training Through Time (OTTT) can train SNNs at each time step and achieve high performance with few time steps (Xiao et al., 2022).

OTTT uses different information for the forward and backward processes. For forwarding, the spike train is used; for backwarding, the weighted summation of spike trains (which we refer to as *spike accumulation*) is used. Therefore, efficient computation on GPUs using the Autograd of libraries such as PyTorch (Paszke et al., 2019) requires operations with spike train and spike accumulation during the forward process (see Fig. 1). Meanwhile, OTTT has the theoretical guarantee that the gradient descent direction is a similar to that of Spike Representations by the weighted firing rate coding (Xiao et al., 2021; Meng et al., 2022). However, these gradients are not shown to be perfectly consistent. To accurately bridge them, it is essential to develop a method that guarantees the gradient can be consistent with each of above two gradients.

In this article, we propose *Spike Accumulation Forwarding* (*SAF*) as a new paradigm for training SNNs. Unlike OTTT, SAF propagates not only backward but also forward processes by spike accumulation (see Fig. 1). By using this process, we can halve the number of operations during the forward process. In addition, because SAF does not require retaining the information of membrane potentials as in Zhou et al. (2021), we can reduce memory usage during training compared to OTTT. Furthermore, this propagation strategy is only executed during training, and during inference, we can replace the propagation with the spike train without approximation error. We demonstrate this by proving that the neurons for spike accumulation are identical to the Leaky-Integrate-and-Fire (LIF) (Stein, 1965) neuron, which is a generalization of the Integrate-and-

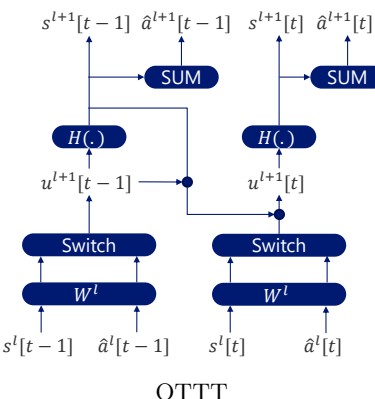
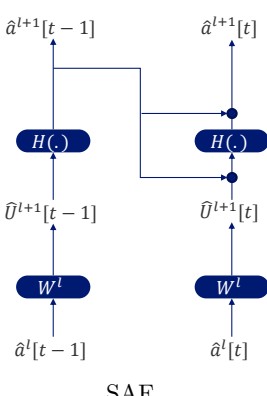

Figure 1: Overview of OTTT and SAF training. OTTT requires operations with spike $\boldsymbol{s}^l[t]$ and spike accumulation $\widehat{\boldsymbol{a}}^l[t]$ during the forward process, while SAF requires operations with $\widehat{\boldsymbol{a}}^l[t]$. Also, $\boldsymbol{u}^l[t]$ represents the membrane potential, and $\widehat{\boldsymbol{U}}^{l+1}[t]$ represents the potential accumulation (see Sec. 4). The SUM layer computes the spike accumulation, and the Switch layer propagates $\boldsymbol{W}^l\boldsymbol{s}$ and $\boldsymbol{W}^l\widehat{\boldsymbol{a}}$ in forwarding and backwarding, respectively. Note that, unlike membrane potentials, the potential accumulation does not require the retention of past information.

Fire (IF) (Lapique, 1907) neuron, which are commonly used as SNN neurons. This result indicates that the SNN composed of LIF neurons can achieve the same accuracy using the trained parameters of SAF (i.e., SAF is capable of inference by the SNN composed of LIF neurons). Furthermore, the gradient of SAF is theoretically consistent with that of the Spike Representation and also with that of OTTT, which trains at each time step. This shows that SAF can accurately bridge the gap between Spike Representation and OTTT, which trains at each time step.

**Main Contributions**

(A) We propose SAF, which trains SNNs by only spike accumulation, halving the number of operations in the forward process, reducing the memory cost, and enabling inference on SNNs composed of LIF neurons.

(B) We prove theoretically that the neurons for spike accumulation are absolutely identical to the LIF neuron.

(C) Our study also shows that the gradient of SAF is theoretically consistent with the gradient of the Spike Representation and also with that of OTTT, which trains each time step.

(D) Brief experiments confirmed that for training at each time step, the training results were in close agreement with OTTT while reducing the training cost.

## 2 RELATED WORK

Regarding SNN training, there are two research directions: conversion from ANN to SNN and direct training. The conversion approach reuses the parameters of the ANN while converting the activation function for the spiking function (Diehl et al., 2015; Deng and Gu, 2020; Han et al., 2020). This approach can be employed by a wide range of many trained deep-learning models, and there are use cases for tasks other than recognition (Kim et al., 2020; Qu et al., 2023). However, because the accuracy tends to be proportional to the number of time steps and although several improvement methods have been proposed (Chowdhury et al., 2021; Wu et al., 2021), high-precision inference is still difficult for small time steps. Meanwhile, direct training does not use the parameters of the trained ANNs. Thus, the non-differentiable SNNs are trained by some approximation techniques. One of the most significant techniques is to utilize the surrogate gradient (SG). SGs enables backpropagation by approximating the gradient of non-differentiable activation functions, and various types of SGs have been proposed (Shrestha and Orchard, 2018; Wu et al., 2018; Lian et al., 2023;

Suetake et al., 2023). Other recently proposed methods include those based on the clamp function (Meng et al., 2022) or implicit differentiation on the equilibrium state (Xiao et al., 2021). These employ Spike Representation, which propagates information differently from the spike train (e.g., firing rate) and has have the advantage of being able to train SNNs like ANNs (Thiele et al., 2019; Zhou et al., 2021). However, because these methods assume $T \to \infty$, $T$ must be large to achieve high accuracy. In addition, these are only differences in how to approximate; the basic approach is similar to SGs. Although there are bio-inspired training methods, such as Hebbian learning rule (Hebb, 2005; Frémaux and Gerstner, 2016) and spike timing dependent plasticity (STBP) (Bi and Poo, 1998; Bengio et al., 2015), we omit them due to the difficulty of training deep models.

In the following, we discuss the OTTT (Xiao et al., 2022) most relevant to our study. Because OTTT is a variant of BPTT with SG, it allows for low-latency training. In addition, it is sufficient for OTTT to maintain the computational graph only for the current time step during training, different from the standard BPTT with SG. Thus, training can be performed with constant memory usage even as time steps increase. However, OTTT requires additional information for propagating the spike accumulation as well as the spike train for the forward process, which can increase training time. In addition, because OTTT is based on the LIF neuron, it must retain the membrane potential, which can increase memory usage. Furthermore, it is important to note that while OTTT and Spike Representation have closely related gradient directions, their gradients do not necessarily match exactly (i.e., the inner product between their gradients is positive).

## 3  Preliminaries

### 3.1  Typical Neuron Model

In this subsection, we explain the LIF neuron, which is widely used in SNNs. The LIF neuron is a neuron model that considers the leakage of the membrane potential, and its discrete computational form is as follows:

$$\begin{cases} \boldsymbol{u}^{l+1}[t] = \lambda(\boldsymbol{u}^{l+1}[t-1] - V_{\text{th}}\,\boldsymbol{s}^{l+1}[t-1]) + \boldsymbol{W}^l \boldsymbol{s}^l[t] + \boldsymbol{b}^{l+1}, \\ \boldsymbol{s}^{l+1}[t] = H(\boldsymbol{u}^{l+1}[t] - V_{\text{th}}), \end{cases} \tag{1}$$

where $\boldsymbol{s}^l[t]$ is the spike train, $\boldsymbol{u}^l[t]$ is the membrane potential, and $\boldsymbol{W}^l$ and $\boldsymbol{b}^l$ are the weight and bias, respectively. $\lambda \leq 1$ is the leaky term, and $\lambda$ is set to 1 if we use the IF neuron, which is a special case of the LIF neuron. Also, $H$ is the element-wise Heaviside step function, that is, $H = 1$ when the membrane potential $\boldsymbol{u}[t]$ exceeds the threshold $V_{\text{th}}$. From this equation, the membrane potentials $\boldsymbol{u}^l[t]$ are computed sequentially and must retain the previous membrane potential $\boldsymbol{u}^l[t-1]$. This is the same in the case of OTTT, which uses the LIF neuron as described below.

### 3.2  Training methods for SNNs

This subsection introduces two training methods that are closely related to our method: Spike Representation and OTTT.

**Spike Representation**

Spike Representation is a method of training SNNs by propagating information differently to the spike trains (Xiao et al., 2021; Meng et al., 2022). In this article, we consider the weighted firing rate $\boldsymbol{a}[t] = \sum_{\tau=0}^{t} \lambda^{t-\tau} \boldsymbol{s}[\tau] / \sum_{\tau=0}^{t} \lambda^{t-\tau}$ as in Xiao et al. (2021); Meng et al. (2022); Xiao et al. (2022). Likewise, we define the weighted average input $\boldsymbol{m}[t] = \sum_{\tau=0}^{t} \lambda^{t-\tau} \boldsymbol{x}[\tau] / \sum_{\tau=0}^{t} \lambda^{t-\tau}$ , where $x$ is the value of the input data. Then, given a convergent sequence $\boldsymbol{m}[t] \to \boldsymbol{m}^*$ $(t \to \infty)$, it is known that $\boldsymbol{a}[t] \to \sigma(\boldsymbol{m}^*/V_{\text{th}})$ $(t \to \infty)$ holds (Xiao et al., 2021), where $\sigma$ is the element-wise clamp function $\sigma(x) = \min(\max(0, x), 1)$. Using this convergence and under the assumption that the time step $T$ is sufficiently large, the weighted firing rate in the $(l+1)$-th layer is approximated as $\boldsymbol{a}^{l+1}[T] \approx \sigma(\boldsymbol{W}^l \boldsymbol{a}^l[T] + \boldsymbol{b}^{l+1}/V_{\text{th}})$. We consider the loss $L$ as $L[t] = \mathcal{L}(\sum_{t=1}^{T} \boldsymbol{s}[t]/T, y)$ (where $\mathcal{L}$ is a convex function like cross-entropy and $y$ is the label). Then the

gradient of $L$ with respect to $\boldsymbol{W}^l$ is computed as follows:

$$\left(\frac{\partial L}{\partial \boldsymbol{W}^l}\right)_{\text{SR}} = \frac{\partial L}{\partial \boldsymbol{a}^N[T]} \left(\prod_{i=N-1}^{l+1} \frac{\partial \boldsymbol{a}^{i+1}[T]}{\partial \boldsymbol{a}^i[T]}\right) \frac{\partial \boldsymbol{a}^{l+1}[T]}{\partial \boldsymbol{W}^l}, \tag{2}$$

where $N$ represents the number of layers.

**Online Training Through Time**

OTTT (Xiao et al., 2022) is a training method based on BPTT with SG. BPTT with SG enables low-latency training; however, during training, it requires the computational graph to be maintained at each time step, resulting in substantial memory usage when a large number of time steps are involved. OTTT solves this problem and allows for training with minimal memory consumption.

In OTTT, for the forward process, (weighted) spike accumulation $\widehat{\boldsymbol{a}}[t] = \sum_{\tau=0}^{t} \lambda^{t-\tau} \boldsymbol{s}[\tau]$ is propagated in addition to spike trains $\boldsymbol{s}[t]$ computed by the LIF neuron. Defining the loss at each time step as $L[t] = \mathcal{L}(\boldsymbol{s}[t], y)/T$, OTTT computes the gradient at time $t$ as follows:

$$\left(\frac{\partial L[t]}{\partial \boldsymbol{W}^l}\right)_{\text{OT}} = \widehat{\boldsymbol{a}}^l[t] \frac{\partial L[t]}{\partial \boldsymbol{s}^N[t]} \left(\prod_{i=N-1}^{l+1} \frac{\partial \boldsymbol{s}^{i+1}[t]}{\partial \boldsymbol{s}^i[t]}\right) \frac{\partial \boldsymbol{s}^{l+1}[t]}{\partial \boldsymbol{u}^{l+1}[t]}. \tag{3}$$

Note that the term $\partial \boldsymbol{s}[t]/\partial \boldsymbol{u}[t]$ is non-differentiable at $u = V_{\text{th}}$; thus, we approximate it with the SG.

Xiao et al. (2022) proposed two types of training approaches: $\text{OTTT}_{\text{O}}$, where parameters are updated at each time step using $\partial L[t]/\partial \boldsymbol{W}^l$, and $\text{OTTT}_{\text{A}}$, where parameters are updated collectively by summing $\partial L[t]/\partial \boldsymbol{W}^l$ up to $T$. In particular, they proved that the gradient descent directions in $\text{OTTT}_{\text{A}}$ and Spike Representation are close, i.e., the inner product between their gradients is positive.

## 4 Spike Accumulation Forwarding

In this section, we introduce our proposed method, SAF, which only propagates (weighted) spike accumulation $\widehat{\boldsymbol{a}}[t]$. We first explain the forward and backward processes. Then, we prove that SAF can be consistent with OTTT and Spike Representation. We also show that the feedback connection can be added to SAF, as in Xiao et al. (2022).

### 4.1 Details of SAF

**Forward process**

As mentioned earlier, for the forward processes of conventional SNNs, the spike trains $\boldsymbol{s}[t]$ are propagated. In other words, the firing state of the spike for each neuron at each time step is retained. In SAF, instead of the spike trains, it propagates (weighted) spike accumulation $\widehat{\boldsymbol{a}}[t] = \sum_{\tau=0}^{t} \lambda^{t-\tau} \boldsymbol{s}[\tau]$, meaning that it retains the (weighted) count of the fired spikes up to the current time for each neuron. Additionally, although in conventional SNNs, the spike firing is determined with the membrane potential $\boldsymbol{u}[t]$, in SAF, it is determined with (weighted) potential accumulation $\widehat{\boldsymbol{U}}[t]$ defined by $\widehat{\boldsymbol{U}}^{l+1}[t] = \lambda \widehat{\boldsymbol{U}}^{l+1}[t-1] + \boldsymbol{W}^l(\widehat{\boldsymbol{a}}^l[t] - \lambda \widehat{\boldsymbol{a}}^l[t-1]) + \boldsymbol{b}^{l+1}$, which corresponds to the membrane potential in the relation (1). With these considerations, SAF is updated as follows:

$$\begin{cases} \widehat{\boldsymbol{U}}^{l+1}[t] = \boldsymbol{W}^l \widehat{\boldsymbol{a}}^l[t] + \boldsymbol{b}^{l+1} \sum_{\tau=0}^{t-1} \lambda^{t-\tau} + \lambda^t \widehat{\boldsymbol{U}}^{l+1}[0], \\ \widehat{\boldsymbol{a}}^{l+1}[t] = \lambda \widehat{\boldsymbol{a}}^{l+1}[t-1] + H(\widehat{\boldsymbol{U}}^{l+1}[t] - V_{\text{th}}(\lambda \widehat{\boldsymbol{a}}^{l+1}[t-1] + 1)), \end{cases} \tag{4}$$

where $\widehat{\boldsymbol{U}}^{l+1}[0]$ is the initial value for potential accumulation, and here, we assume it to be the initial membrane potential $\boldsymbol{u}^{l+1}[0]$. Here, the membrane potential $\boldsymbol{u}^{l+1}[t]$ and spike trains $\boldsymbol{s}^{l+1}[t]$ in the LIF model can be expressed by $\widehat{\boldsymbol{U}}^{l+1}[t]$ and $\widehat{\boldsymbol{a}}^{l+1}[t-1]$ as follows:

$$\begin{cases} \boldsymbol{u}^{l+1}[t] = \widehat{\boldsymbol{U}}^{l+1}[t] - V_{\text{th}} \lambda \widehat{\boldsymbol{a}}^{l+1}[t-1], \\ \boldsymbol{s}^{l+1}[t] = H(\widehat{\boldsymbol{U}}^{l+1}[t] - V_{\text{th}}(\lambda \widehat{\boldsymbol{a}}^{l+1}[t-1] + 1)). \end{cases} \tag{5}$$

For derivations of (4) and (5), refer to Appendix A.1. Note that, as shown in (4), SAF does not need to retain the past potential accumulation $\widehat{U}^{l+1}[t-1]$. Meanwhile, the various SNNs, including OTTT, require the LIF neurons used for training to retain the previous membrane potentials $\boldsymbol{u}^{l+1}[t-1]$, as described above. Therefore, SAF can reduce the memory usage for the forward process compared to OTTT.

As a result, it becomes possible to compute $\boldsymbol{u}^{l+1}[t]$ and $\boldsymbol{s}^{l+1}[t]$ during the process of obtaining $\widehat{U}^{l+1}[t]$ and $\widehat{\boldsymbol{a}}^{l+1}[t]$. Because it is possible to compute $\widehat{U}^{l+1}[t]$ and $\widehat{\boldsymbol{a}}^{l+1}[t]$ from $\boldsymbol{u}^{l+1}[t]$ and $\boldsymbol{s}^{l+1}[\tau]$ ($\tau = 1, \ldots, t$), the forward processes of SAF and SNN composed of LIF neurons are mutually convertible. Additionally, because the IF neuron is a special case of an LIF neuron (i.e., $\lambda = 1$), the forward processes of SAF, when $\lambda = 1$, and SNN composed of IF neurons are mutually convertible. Furthermore, in OTTT, both $\boldsymbol{s}[t]$ and $\widehat{\boldsymbol{a}}[t]$ need to be propagated during the forward process for efficient GPU computation, whereas in SAF, only $\widehat{\boldsymbol{a}}[t]$ needs to be propagated (see Fig. 1). Therefore, SAF can reduce the computation time during training.

**Backward process**

As with OTTT, SAF can be trained in two different ways. The first method updates the parameters by computing the gradient at each time step. We call this *SAF-E*. Let $L_E[t] = \mathcal{L}(\boldsymbol{s}^N[t], \boldsymbol{y})/T$ be the loss function. Assuming that $L_E[t]$ depends only on $\widehat{\boldsymbol{a}}^l[t]$ and $\widehat{U}^l[t]$, i.e., not on anything up to $t-1$, we calculate the derivative based on the definition of forward propagation as

$$\frac{\partial L_E[t]}{\partial \boldsymbol{W}^l} = \widehat{\boldsymbol{a}}^l[t] \frac{\partial L_E[t]}{\partial \widehat{\boldsymbol{a}}^N[t]} \left( \prod_{i=N-1}^{l+1} \frac{\partial \widehat{\boldsymbol{a}}^{i+1}[t]}{\partial \widehat{\boldsymbol{a}}^i[t]} \right) \frac{\partial \widehat{\boldsymbol{a}}^{l+1}[t]}{\partial \widehat{U}^{l+1}[t]}.$$

Note that $\partial \widehat{\boldsymbol{a}}^{l+1}[t]/\partial \widehat{U}^{l+1}[t]$ is non-differentiable; we approximate it with SG. Detailed calculations are given in Appendix A.2. Here, we set

$$\boldsymbol{g}_{\widehat{U}}^{l+1}[t] = \frac{\partial L_E[t]}{\partial \widehat{\boldsymbol{a}}^N[t]} \left( \prod_{i=N-1}^{l+1} \frac{\partial \widehat{\boldsymbol{a}}^{i+1}[t]}{\partial \widehat{\boldsymbol{a}}^i[t]} \right) \frac{\partial \widehat{\boldsymbol{a}}^{l+1}[t]}{\partial \widehat{U}^{l+1}[t]}.$$

Then, it hold that

$$\frac{\partial L_E[t]}{\partial \boldsymbol{W}^l} = \widehat{\boldsymbol{a}}^l[t] \, \boldsymbol{g}_{\widehat{U}}^{l+1}[t]. \tag{6}$$

The second method calculates the gradient only at the final time step and updates the parameters. We call this *SAF-F*. Let $L_F = \mathcal{L}(\sum_{t=0}^{T} \lambda^{T-t} \boldsymbol{s}^N[t] / \sum_{t=0}^{T} \lambda^{T-t}, \boldsymbol{y})$ be a loss function. As with SAF-E, suppose that $L_F$ depends only on $\widehat{\boldsymbol{a}}^l[T]$ and $\widehat{U}^l[T]$. Simply replacing $t$ with $T$ and $L_E$ with $L_F$ in the above calculation, we obtain

$$\frac{\partial L_F}{\partial \boldsymbol{W}^l} = \widehat{\boldsymbol{a}}^l[T] \, \boldsymbol{g}_{\widehat{U}}^{l+1}[T]. \tag{7}$$

## 4.2 Equivalence with OTTT$_\text{O}$ and Spike Representation

In this subsection, we show that SAF-E is equivalent to OTTT$_\text{O}$ and SAF-F is equivalent to Spike Representation, i.e., the forward and backward processes are consistent, respectively. This means that we can train SNNs by SAF and infer by LIF neurons.

**Equivalence with OTTT$_\text{O}$**

We will transform the gradient of SAF-E to be consistent with that of OTTT$_\text{O}$ when the loss function is $L_E[t]$. Because $L_E[t]$ does not include any argument before $t$, we obtain

$$\frac{\partial L_E[t]}{\partial \widehat{\boldsymbol{a}}^N[t]} = \frac{\partial L_E[t]}{\partial \boldsymbol{s}^N[t]}. \tag{8}$$

The following two equations hold from the forward processes of SAF and OTTT:

$$\frac{\partial \widehat{\boldsymbol{a}}^{i+1}[t]}{\partial \widehat{\boldsymbol{a}}^i[t]} = \frac{\partial \boldsymbol{s}^{i+1}[t]}{\partial \boldsymbol{s}^i[t]}, \quad \frac{\partial \widehat{\boldsymbol{a}}^{l+1}[t]}{\partial \widehat{U}^{l+1}[t]} = \frac{\partial \boldsymbol{s}^{l+1}[t]}{\partial \boldsymbol{u}^{l+1}[t]}. \tag{9}$$

By substituting (8), (9) for $g_{\widehat{U}}^{l+1}$ in (6), we have

$$g_{\widehat{U}}^{l+1}[t] = \frac{\partial L_E[t]}{\partial s^N[t]} \left( \prod_{i=N-1}^{l+1} \frac{\partial s^{i+1}[t]}{\partial s^i[t]} \right) \frac{\partial s^{l+1}[t]}{\partial u^{l+1}[t]}.$$

Hence, the following theorem holds from (3).

**Theorem 1.** *The backward processes of SAF-E and OTTT$_{\mathrm{O}}$ are identical, that is,* $\dfrac{\partial L_E[t]}{\partial \boldsymbol{W}^l} = \left( \dfrac{\partial L_E[t]}{\partial \boldsymbol{W}^l} \right)_{\mathrm{OT}}.$

A detailed proof is given in Appendix A.3. Because we have already confirmed that the forward process is consistent, SAF-E and OTTT$_{\mathrm{O}}$ are equivalent.

**Equivalence with Spike Representation**

Now, we show that SAF-F is equivalent to Spike Representation. Setting the loss function of Spike Representation as $L_F$, from the expression (2), we have

$$\left( \frac{\partial L_F}{\partial \boldsymbol{W}^l} \right)_{\mathrm{SR}} = \frac{\widehat{\boldsymbol{a}}^l[T]}{V_{\mathrm{th}}} \left( \frac{\partial L}{\partial \widehat{\boldsymbol{a}}^N[T]} \left( \prod_{i=N-1}^{l+1} \frac{\partial \widehat{\boldsymbol{a}}^{i+1}[T]}{\partial \widehat{\boldsymbol{a}}^i[T]} \right) \odot \boldsymbol{d}^{l+1}[T]^\top \right),$$

where $\boldsymbol{d}^{l+1}[T] = \sigma'\left( (\boldsymbol{W}^l \widehat{\boldsymbol{a}}^l[T]/\Lambda + \boldsymbol{b}^{l+1})/V_{\mathrm{th}} \right)$, $\Lambda = \sum_{\tau=0}^T \lambda^{T-\tau}$, and $\odot$ is the element-wise product. Now we assume that

$$\frac{\partial \boldsymbol{s}^{l+1}[T]}{\partial \boldsymbol{u}^{l+1}[T]} = \mathrm{diag}(\boldsymbol{d}^{l+1}[T]),$$

for any $l = 0, \ldots, N-1$, where $\mathrm{diag}(\boldsymbol{d}^{l+1}[T])$ is a diagonal matrix constructed from $\boldsymbol{d}^{l+1}[T]$. The reason why this assumption is valid discussed in Xiao et al. (2022). Then, we obtain

$$\left( \frac{\partial L_F}{\partial \boldsymbol{W}^l} \right)_{\mathrm{SR}} = \frac{1}{V_{\mathrm{th}}} \widehat{\boldsymbol{a}}^l[T] \, \boldsymbol{g}_{\widehat{U}}^{l+1}[T].$$

Hence, the following theorem holds.

**Theorem 2.** *Suppose that $\boldsymbol{m}[t]$ converges when $t \to \infty$. Then, for sufficiently large $T$, the backward processes of SAF-F and Spike Representation are essentially identical, that is,* $\dfrac{\partial L_F}{\partial \boldsymbol{W}^l} = V_{\mathrm{th}} \left( \dfrac{\partial L_F}{\partial \boldsymbol{W}^l} \right)_{\mathrm{SR}}.$

See Appendix A.4 for the complete proof. Because we have already confirmed that the forward process is consistent, SAF-F and OTTT$_{\mathrm{A}}$ are equivalent.

### 4.3 FEEDBACK CONNECTION

In this subsection, we will consider SAF, including the feedback connections, which have been frequently used in recent years (Xiao et al., 2021; 2022; 2023). Note that SAF with feedforward connections can be discussed in the same way, and details are provided in Appendix B.1.

To begin, we consider the SNN composed of LIF neurons. The forward process of the $(q+1)$-th layer of the SNN with a feedback connection from the $p$-th layer to the $(q+1)$-th layer (where $q < p$) with weight $\boldsymbol{W}_b$ is as follows:

$$\begin{cases} \boldsymbol{u}^{q+1}[t] = \lambda(\boldsymbol{u}^{q+1}[t-1] - V_{\mathrm{th}} \, \boldsymbol{s}^{q+1}[t-1]) + \boldsymbol{W}^q \boldsymbol{s}^q[t] + \boldsymbol{b}^{q+1} + \boldsymbol{W}_b \, \boldsymbol{s}^p[t-1], \\ \boldsymbol{s}^{q+1}[t] = H(\boldsymbol{u}^{q+1}[t] - V_{\mathrm{th}}). \end{cases}$$

Note that the layers other than the $(q+1)$-th layer are the same as in (1). Meanwhile, the forward process of the $(q+1)$-th layer of SAF is as follows:

$$\begin{cases} \widehat{\boldsymbol{U}}^{q+1}[t] = \boldsymbol{W}^q \widehat{\boldsymbol{a}}^q[t] + \boldsymbol{b}^{q+1}(\sum_{\tau=0}^{t-1} \lambda^\tau) + \lambda^t \widehat{\boldsymbol{U}}^{q+1}[0] + \boldsymbol{W}_b \, \widehat{\boldsymbol{a}}^p[t-1], \\ \widehat{\boldsymbol{a}}^{q+1}[t] = \widehat{\boldsymbol{a}}^{q+1}[t-1] + H(\widehat{\boldsymbol{U}}^{q+1} - V_{\mathrm{th}}(\lambda \widehat{\boldsymbol{a}}^{q+1}[t-1] + 1)). \end{cases}$$

As with LIF neurons, the layers other than the $(q+1)$-th layer are the same as in (4). The forward processes of SAF and LIF with feedback connection are mutually convertible.

Regarding the backward processes, the gradients for parameters other than $\boldsymbol{W}_b$ are the same as when there is no feedback connection. The derivative with respect to $\boldsymbol{W}_b$ is caluclated as

$$\frac{\partial L_E[t]}{\partial \boldsymbol{W}_b} = \widehat{\boldsymbol{a}}^p[t-1] \frac{\partial L_E[t]}{\partial \widehat{\boldsymbol{a}}^N[t]} \left( \prod_{i=N-1}^{q+1} \frac{\partial \widehat{\boldsymbol{a}}^{i+1}[t]}{\partial \widehat{\boldsymbol{a}}^i[t]} \right) \frac{\partial \widehat{\boldsymbol{a}}^{q+1}[t]}{\partial \widehat{\boldsymbol{U}}^{q+1}[t]}.$$

Therefore, we obtain $\partial L_E[t]/\partial \boldsymbol{W}_b = \widehat{\boldsymbol{a}}^p[t-1]\,\boldsymbol{g}_{\widehat{\boldsymbol{U}}}^{q+1}[t]$ for SAF-E and $\partial L_F/\partial \boldsymbol{W}_b = \widehat{\boldsymbol{a}}^p[T-1]\,\boldsymbol{g}_{\widehat{\boldsymbol{U}}}^{q+1}[T]$ for SAF-F. Thus, it can be confirmed that SAF-E is equivalent to $\text{OTTT}_O$. Also, the gradient descent directions in SAF-F and Spike Representation are close. For the proof, refer to Appendix B.2

## 5 EXPERIMENTS

In Sec. 4.2, we theoretically proved that SAF-E and $\text{OTTT}_O$ as well as SAF-F and spike representation are equivalent. In this section, we experimentally compare these methods. As complex and large datasets make it difficult to analyze the results, we trained SAF on the CIFAR-10 dataset (Krizhevsky and Hinton, 2009) and inferred with SNN composed of LIF neurons. This experiment was performed five times with different initial parameters, and all approximation was executed by the sigmoid-like SG for fair comparison. We used the same experimental setup as (Xiao et al., 2022), including the choice of SG. The code was written in PyTorch (Paszke et al., 2019), and the experiments were executed using one GPU, an NVIDIA A10. The implementation details and pseudocode are in Appendixes C and D. The main objective here is to analyze whether there are any inconsistencies between theory and experiment rather than to achieve state-of-the-art performance.

### 5.1 ANALYSIS OF SAF-E

We experimentally analyze the performance of SAF-E. First, we compare accuracy. Table 1 shows the accuracy when we set $T = 6$. Note that the values in parentheses are the changes in accuracy due to inference by the SNN composed of LIF neurons. As shown in this table, the accuracies of SAF-E and $\text{OTTT}_O$ are almost identical, and the accuracy change due to inference with SNNs consisting of LIF neurons is also almost negligible. Figure 2 shows the accuracy and loss curves during training. This indicates that not only the results but also the progress during training are comparable. Therefore, we confirmed experimentally that SAF-E and $\text{OTTT}_O$ are equivalent, as mentioned in Sec. 4.2. Similar results were obtained when there was a feedback connection (see Appendix E ).

Next, we compare the training costs. From Table 1, it can be seen that SAF-E takes less time to train and uses less memory during training than $\text{OTTT}_O$. However, OTTT can be executed with constant memory usage even as time steps increase. Therefore, we compared the training time and memory usage at different time steps. Figures 3 (A) and (B) show the training time and memory at different time steps. Note that the training time was measured in one batch. It can be seen that the memory usage of SAF-E does not increase even if the number of time steps increases, similar to $\text{OTTT}_O$. Also, from Fig. 3 (B), we can see that SAF-E uses less memory than $\text{OTTT}_O$. This result stems from the fact that SAF does not need to maintain the previous membrane potential.

Finally, we compare the firing rate. As shown in Table 1, the total firing rates of SAF-E and $\text{OTTT}_O$ are almost the same. Also, the amount of change due to inference with SNNs consisting of LIF neurons is also almost negligible, similar to the case for accuracy. Furthermore, from Fig. 3 (C), the firing rates of each layer are almost identical too.

These results indicate that using SAF-E can reduce the training time and memory compared to $\text{OTTT}_O$ while achieving almost the same firing rate and accuracy. It was also shown that using the parameters trained with SAF-E to infer with the SNN consisting of LIF neurons is feasible.

Table 1: Performance comparison of SAF-E and $\text{OTTT}_\text{O}$. The values in parentheses were the changes in accuracy and total firing rate due to inference by the SNN composed of LIF neurons. Note that training times were measured in one minibatch, and training time and memory were not perturbed between trials.

| Method | Memory [GB] | Training Time [sec] | Firing rate [%] | Accuracy [%] |
|---|---|---|---|---|
| $\text{OTTT}_\text{O}$ | 1.330 | 0.351 | 15.14±0.17 | 93.44±0.15 |
| SAF-E (ours) | 0.921 | 0.246 | 14.76±0.15 ($1.048\times10^{-5}$) | 93.54±0.17 (0.016) |

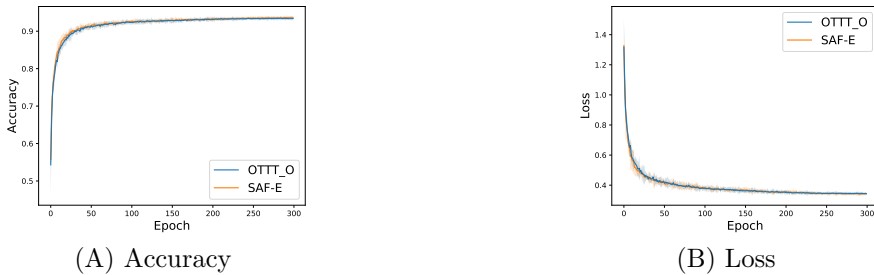

(A) Accuracy         (B) Loss

Figure 2: Accuracy and loss curves of SAF-E and $\text{OTTT}_\text{O}$. It can be seen that the curves overlap, which agree with the fact that SAF-E is theoretically consistent with $\text{OTTT}_\text{O}$.

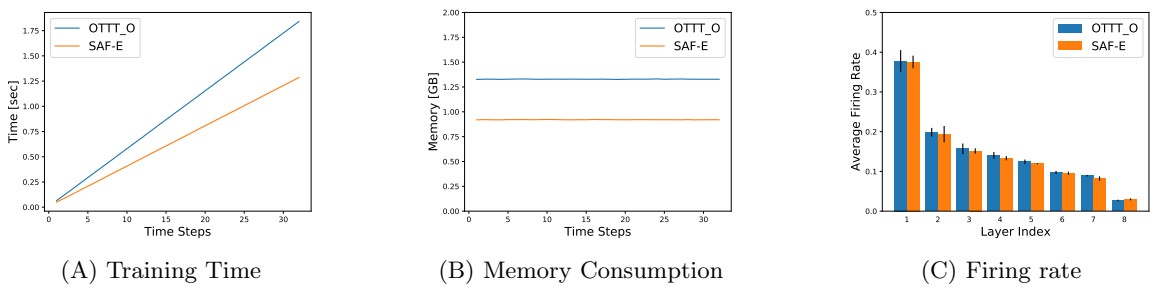

(A) Training Time      (B) Memory Consumption      (C) Firing rate

Figure 3: Training time, memory consumption, and firing rate of each layer of SAF-E and $\text{OTTT}_\text{O}$.

## 5.2 Analysis of SAF-F

In this subsection, we experimentally analyse the performance of SAF-F. First, we compare accuracy. The results are shown in Table 2 and Fig. 4. Note that Spike Representation methods, except for SAF-F, do not use SGs, which makes precise comparisons difficult. Therefore, we only compared SAF-F with $\text{OTTT}_\text{A}$. As with the previous results, the accuracy change due to inference with SNNs consisting of LIF neurons is almost negligible. Meanwhile, the accuracies of SAF-F and $\text{OTTT}_\text{O}$ are close, though from the perspective of standard deviation, there seems to be a difference. From Sec. 4.2, the gradient directions of Spike Representation and SAF-F are identical, but Spike Representation and $\text{OTTT}_\text{A}$ are only close. Therefore, SAF-F and $\text{OTTT}_\text{A}$ are also only close. This is thought to be the cause of the differences in accuracy and loss.

Next, we compare the training costs. From Table 2, it can be seen that SAF-F requires less time for training and uses less memory than $\text{OTTT}_\text{A}$. This trend is also similar when the time step is varied (see Figs. 5 (A) and (B)).

Finally, we compare firing rate. As shown in Table 2 as for accuracy, the change of the total firing rate by inferring with SNNs consisting of LIF neurons is almost negligible. Meanwhile, the total firing rate of SAF-F is smaller than of $\text{OTTT}_\text{A}$. In addition, from Fig. 5 (C), it can be seen that the firing rate of each layer (especially the first layer) is smaller in SAF-F than $\text{OTTT}_\text{A}$. These differences also indicate that SAF-F and $\text{OTTT}_\text{A}$ are generally not identical.

Table 2: Performance comparison of SAF-F and OTTT$_\text{A}$. The values in parentheses are the changes in accuracy and total firing rate due to inference by the SNN composed of LIF neurons. Note that training times were measured in one minibatch, and training time and memory were not perturbed between trials.

| Method | Memory [GB] | Training Time [sec] | Firing rate [%] | Accuracy [%] |
|---|---|---|---|---|
| OTTT$_\text{A}$ | 1.330 | 0.348 | 15.51±0.10 | 93.39±0.16 |
| SAF-F (ours) | 0.893 | 0.127 | 10.50±0.19 ($3.306\times10^{-5}$) | 93.09±0.15 (0.076) |

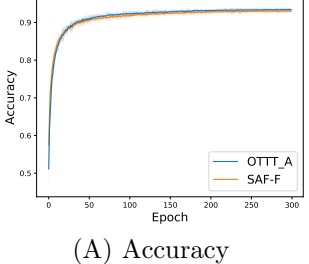

(A) Accuracy

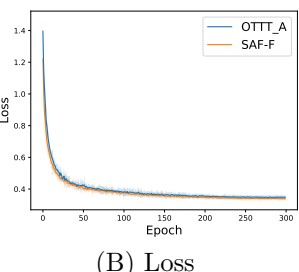

(B) Loss

Figure 4: Accuracy and loss curves of SAF-F and OTTT$_\text{A}$. It can be seen that the curves do not overlap slightly, which agree with the theoretical result that SAF-F is close to (but not always consistent with) OTTT$_\text{A}$.

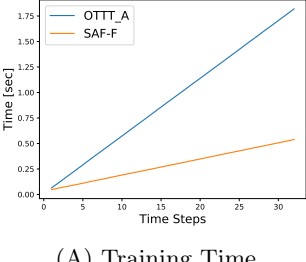

(A) Training Time

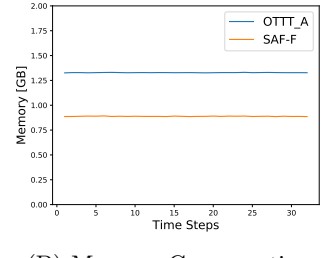

(B) Memory Consumption

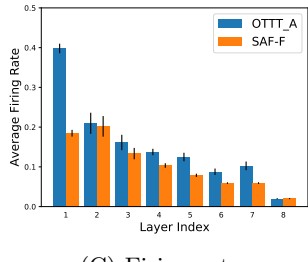

(C) Firing rate

Figure 5: Training time, memory consumption, and firing rate of each layer of SAF-F and OTTT$_\text{A}$. It can be seen that the firing rates do not match, which agree with the theoretical result that SAF-F is not always consistent with OTTT$_\text{A}$.

From the above analysis, we can say that SAF-F is a better choice than OTTT$_\text{A}$ in terms of training time, memory usage, and firing rate. Also, as with SAF-E, inference can be performed by the standard SNN using the parameters trained by SAF-F.

## 6 Conclusion and Future Work

This article proposed SAF. SAF is a training method of SNNs that propagates the spike accumulation during training; however, SAF propagates the spike trains during inference, as do other SNNs. This article showed that SAF trained at each time step (SAF-E) is equivalent to OTTT$_\text{O}$, and SAF trained at the final time step (SAF-F) is identical to Spike Representation. We conducted experiments on the CIFAR-10 dataset and confirmed that the experimental results are consistent with these assertions and that training time and memory are reduced compared to OTTT.

SAF presented in this article assumes training on a GPU. Therefore, it may not be suitable for training on neuromorphic chips. However, we believe executing training on GPUs and inference on neuromorphic chips is reasonable. We plan to verify inference on a neuromorphic chip in the future.

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

# Appendix

## A  DERIVATION AND PROOFS

### A.1  DERIVATION OF (4) AND (5)

In this subsection, we derive (4) and (5) through proving that (4) and (5) hold for any $t \in \{1, \dots, T\}$ with mathematical induction.

First, we prove that (4) and (5) hold for $t = 1$. We compute $\widehat{\boldsymbol{U}}^{l+1}[1]$ based on definition as follows:

$$\widehat{\boldsymbol{U}}^{l+1}[1] = \lambda \widehat{\boldsymbol{U}}^{l+1}[0] + \boldsymbol{W}^l(\widehat{\boldsymbol{a}}^l[1] - \lambda \widehat{\boldsymbol{a}}^l[0]) + \boldsymbol{b^{l+1}}$$
$$= \boldsymbol{W}^l \widehat{\boldsymbol{a}}^l[1] + \boldsymbol{b^{l+1}} + \lambda \widehat{\boldsymbol{U}}^{l+1}[0].$$

Taking account into $\widehat{\boldsymbol{a}}^{l+1}[0] = \widehat{\boldsymbol{s}}^{l+1}[0] = 0$, we obtain

$$\boldsymbol{u}^{l+1}[1] = \lambda(\boldsymbol{u}^{l+1}[0] - V_{\text{th}}\,\boldsymbol{s}^{l+1}[0]) + \boldsymbol{W}^l \boldsymbol{s}^l[1] + \boldsymbol{b^{l+1}}$$
$$= \widehat{\boldsymbol{U}}^{l+1}[0] - V_{\text{th}}\,\widehat{\boldsymbol{a}}^{l+1}[0]) + \boldsymbol{W}^l(\widehat{\boldsymbol{a}}^l[1] - \lambda \widehat{\boldsymbol{a}}^l[0]) + \boldsymbol{b^{l+1}}$$
$$= \widehat{\boldsymbol{U}}^{l+1}[1] - V_{\text{th}}\,\widehat{\boldsymbol{a}}^{l+1}[0].$$

With this equation, we have

$$\boldsymbol{s}^{l+1}[1] = H(\boldsymbol{u}^{l+1}[1] - V_{\text{th}})$$
$$= H(\widehat{\boldsymbol{U}}^{l+1}[1] - V_{\text{th}}(\lambda \widehat{\boldsymbol{a}}^{l+1}[0] + 1)).$$

Then, $\widehat{\boldsymbol{a}}^{l+1}[1]$ can be computed as follows:

$$\widehat{\boldsymbol{a}}^{l+1}[1] = \sum_{\tau=0}^{1} \lambda^{1-\tau} \boldsymbol{s}^{l+1}[\tau]$$
$$= \lambda \widehat{\boldsymbol{a}}^{l+1}[0] + \boldsymbol{s}^{l+1}[1]$$
$$= \lambda \widehat{\boldsymbol{a}}^{l+1}[0] + H(\widehat{\boldsymbol{U}}^{l+1}[1] - V_{\text{th}}(\lambda \widehat{\boldsymbol{a}}^{l+1}[0] + 1)).$$

Therefore, (4) and (5) hold for $t = 1$.

Next, we prove that (4) and (5) hold for $t$ when they hold for any $\tau \in \{1, \dots, t-1\}$. Assuming that (4) and (5) hold for any $\tau \in \{1, \dots, t-1\}$, we have

$$\widehat{\boldsymbol{U}}^{l+1}[t] = \lambda \widehat{\boldsymbol{U}}^{l+1}[t-1] + \boldsymbol{W}^l(\widehat{\boldsymbol{a}}^l[t] - \lambda \widehat{\boldsymbol{a}}^l[t-1]) + \boldsymbol{b^{l+1}}$$
$$= \lambda \left( \boldsymbol{W}^l \widehat{\boldsymbol{a}}^l[t-1] + \boldsymbol{b^{l+1}} \sum_{\tau=0}^{t-2} \lambda^{t-\tau} + \lambda^{t-1} \widehat{\boldsymbol{U}}^{l+1}[0] \right) + \boldsymbol{W}^l(\widehat{\boldsymbol{a}}^l[t] - \lambda \widehat{\boldsymbol{a}}^l[t-1]) + \boldsymbol{b^{l+1}}$$
$$= \boldsymbol{W}^l \widehat{\boldsymbol{a}}^l[t] + \boldsymbol{b^{l+1}} \sum_{\tau=0}^{t-1} \lambda^{t-\tau} + \lambda^t \widehat{\boldsymbol{U}}^{l+1}[0].$$

Also, $\boldsymbol{u}^{l+1}[t]$ is computed as follows:

$$\boldsymbol{u}^{l+1}[t] = \lambda(\boldsymbol{u}^{l+1}[t-1] - V_{\text{th}}\,\boldsymbol{s}^{l+1}[t-1]) + \boldsymbol{W}^l \boldsymbol{s}^l[t] + \boldsymbol{b^{l+1}}$$
$$= \lambda(\widehat{\boldsymbol{U}}^{l+1}[t-1] - V_{\text{th}}(\lambda \widehat{\boldsymbol{a}}^{l+1}[t-2] + \boldsymbol{s}^{l+1}[t-1])) + \boldsymbol{W}^l(\widehat{\boldsymbol{a}}^l[t] - \lambda \widehat{\boldsymbol{a}}^l[t-1]) + \boldsymbol{b^{l+1}}$$
$$= \widehat{\boldsymbol{U}}^{l+1}[t] - V_{\text{th}}\lambda \widehat{\boldsymbol{a}}^{l+1}[t-1].$$

With this equation, we have

$$\boldsymbol{s}^{l+1}[t] = H(\boldsymbol{u}^{l+1}[t] - V_{\text{th}})$$
$$= H(\widehat{\boldsymbol{U}}^{l+1}[t] - V_{\text{th}}(\lambda \widehat{\boldsymbol{a}}^{l+1}[t-1] + 1)).$$

Then, $\widehat{\boldsymbol{a}}^{l+1}[t]$ can be computed as follows:

$$\widehat{\boldsymbol{a}}^{l+1}[t] = \sum_{\tau=0}^{t} \lambda^{t-\tau} \boldsymbol{s}^{l+1}[\tau]$$
$$= \lambda \widehat{\boldsymbol{a}}^{l+1}[t-1] + \boldsymbol{s}^{l+1}[t]$$
$$= \lambda \widehat{\boldsymbol{a}}^{l+1}[t-1] + H(\widehat{\boldsymbol{U}}^{l+1}[t] - V_{\mathrm{th}}(\lambda \widehat{\boldsymbol{a}}^{l+1}[t-1] + 1)).$$

Hence, (4) and (5) hold for $t$ when they hold for any $\tau \in \{1, \ldots, t-1\}$.

Therefore, (4) and (5) hold any $t \in \{1, \ldots, T\}$.

## A.2 DERIVATION OF (6) AND (7)

First, since $L_E[t] = \mathcal{L}(\boldsymbol{s}^N[t], \boldsymbol{y})/T$, we have

$$\frac{\partial L_E[t]}{\partial \boldsymbol{W}^l} = \frac{\partial L_E[t]}{\partial \boldsymbol{s}^N[t]} \frac{\partial \boldsymbol{s}^N[t]}{\partial \boldsymbol{W}^l}.$$

Assuming that $L_E[t]$ depends only on $\widehat{\boldsymbol{a}}^l[t]$ and $\widehat{\boldsymbol{U}}^l[t]$, i.e., not on anything up to $t-1$, we regard $\widehat{\boldsymbol{a}}^N[t]$ as $\boldsymbol{s}^N[t] + \text{Const.}$ Then, we calculate that

$$\frac{\partial L_E[t]}{\partial \boldsymbol{W}^l} = \frac{\partial L_E[t]}{\partial \boldsymbol{s}^N[t]} \frac{\partial \boldsymbol{s}^N[t]}{\partial \widehat{\boldsymbol{a}}^N[t]} \frac{\partial \widehat{\boldsymbol{a}}^N[t]}{\partial \boldsymbol{W}^l} = \frac{\partial L_E[t]}{\partial \widehat{\boldsymbol{a}}^N[t]} \frac{\partial \widehat{\boldsymbol{a}}^N[t]}{\partial \boldsymbol{W}^l}. \tag{10}$$

Next, it follows from (4) that

$$\frac{\partial \widehat{\boldsymbol{a}}^N[t]}{\partial \boldsymbol{W}^l} = \frac{\partial \widehat{\boldsymbol{a}}^N[t]}{\partial \widehat{\boldsymbol{a}}^N[t-1]} \frac{\partial \widehat{\boldsymbol{a}}^N[t-1]}{\partial \boldsymbol{W}^l} + \frac{\partial \widehat{\boldsymbol{a}}^N[t]}{\partial \widehat{\boldsymbol{U}}^N[t]} \frac{\partial \widehat{\boldsymbol{U}}^N[t]}{\partial \boldsymbol{W}^l}$$

$$= \frac{\partial \widehat{\boldsymbol{a}}^N[t]}{\partial \widehat{\boldsymbol{U}}^N[t]} \frac{\partial \widehat{\boldsymbol{U}}^N[t]}{\partial \widehat{\boldsymbol{a}}^{N-1}[t]} \frac{\partial \widehat{\boldsymbol{a}}^{N-1}[t]}{\partial \boldsymbol{W}^l} \tag{11}$$

$$= \frac{\partial \widehat{\boldsymbol{a}}^N[t]}{\partial \widehat{\boldsymbol{a}}^{N-1}[t]} \frac{\partial \widehat{\boldsymbol{a}}^{N-1}[t]}{\partial \boldsymbol{W}^l}, \tag{12}$$

where we regard $\partial \widehat{\boldsymbol{a}}^N[t-1]/\partial \boldsymbol{W}^l$ as 0, and $\widehat{\boldsymbol{U}}^N[t]$ as a function of $\widehat{\boldsymbol{a}}^{N-1}[t]$.

By repeating the process from (11) to (12) in the same way, we derive

$$\frac{\partial \widehat{\boldsymbol{a}}^N[t]}{\partial \boldsymbol{W}^l} = \left( \prod_{i=N-1}^{l+1} \frac{\partial \widehat{\boldsymbol{a}}^{i+1}[t]}{\partial \widehat{\boldsymbol{a}}^i[t]} \right) \frac{\partial \widehat{\boldsymbol{a}}^{l+1}[t]}{\partial \boldsymbol{W}^l}.$$

Therefore, the gradient of SAF-E is calculated as follows:

$$\frac{\partial L_E[t]}{\partial \boldsymbol{W}^l} = \frac{\partial L_E[t]}{\partial \widehat{\boldsymbol{a}}^N[t]} \left( \prod_{i=N-1}^{l+1} \frac{\partial \widehat{\boldsymbol{a}}^{i+1}[t]}{\partial \widehat{\boldsymbol{a}}^i[t]} \right) \frac{\partial \widehat{\boldsymbol{a}}^{l+1}[t]}{\partial \boldsymbol{W}^l}$$

$$= \widehat{\boldsymbol{a}}^l[t] \frac{\partial L_E[t]}{\partial \widehat{\boldsymbol{a}}^N[t]} \left( \prod_{i=N-1}^{l+1} \frac{\partial \widehat{\boldsymbol{a}}^{i+1}[t]}{\partial \widehat{\boldsymbol{a}}^i[t]} \right) \frac{\partial \widehat{\boldsymbol{a}}^{l+1}[t]}{\partial \widehat{\boldsymbol{U}}^{l+1}[t]}, \tag{13}$$

where note that the Heaviside step function $H$ is element-wise. This concludes the derivation of (6) by setting

$$\boldsymbol{g}_{\widehat{U}}^{l+1}[t] = \frac{\partial L_E[t]}{\partial \widehat{\boldsymbol{a}}^N[t]} \left( \prod_{i=N-1}^{l+1} \frac{\partial \widehat{\boldsymbol{a}}^{i+1}[t]}{\partial \widehat{\boldsymbol{a}}^i[t]} \right) \frac{\partial \widehat{\boldsymbol{a}}^{l+1}[t]}{\partial \widehat{\boldsymbol{U}}^{l+1}[t]}.$$

Derivation of (7) is almost the same as that of (6). As with SAF-E, suppose that $L_F$ depends only on $\widehat{\boldsymbol{a}}^l[T]$ and $\widehat{\boldsymbol{U}}^l[T]$. Since $L_F = \mathcal{L}(\widehat{\boldsymbol{a}}^N[t]/\sum_{t=0}^T \lambda^{T-t}, \boldsymbol{y})$, (10) holds when $t$ is replaced with $T$, and $L_E[t]$ is replaced with $L_F$. The rest derivation is the same as in case (6). Then we have

$$\frac{\partial L_F}{\partial \boldsymbol{W}^l} = \widehat{\boldsymbol{a}}^l[T] \frac{\partial L_F}{\partial \widehat{\boldsymbol{a}}^N[T]} \left( \prod_{i=N-1}^{l+1} \frac{\partial \widehat{\boldsymbol{a}}^{i+1}[T]}{\partial \widehat{\boldsymbol{a}}^i[T]} \right) \frac{\partial \widehat{\boldsymbol{a}}^{l+1}[T]}{\partial \widehat{\boldsymbol{U}}^{l+1}[T]} = \widehat{\boldsymbol{a}}^l[T] \, \boldsymbol{g}_{\widehat{\boldsymbol{U}}}^{l+1}[T]. \tag{14}$$

Note that $\partial \widehat{\boldsymbol{a}}^{l+1}[t]/\partial \widehat{\boldsymbol{U}}^{l+1}[t]$ in (13) and (14) is non-differentiable; we approximate it with the SG.

### A.3 PROOF OF THEOREM 1

**Theorem 1.** *The backward processes of SAF-E and OTTT$_O$ are identical , that is,* $\dfrac{\partial L_E[t]}{\partial \boldsymbol{W}^l} = \left( \dfrac{\partial L_E[t]}{\partial \boldsymbol{W}^l} \right)_{\text{OT}}$.

*Proof.* The gradient of SAF-E and OTTT$_O$ are as follows:

$$\frac{\partial L_E[t]}{\partial \boldsymbol{W}^l} = \widehat{\boldsymbol{a}}^l[t] \, \boldsymbol{g}_{\widehat{\boldsymbol{U}}}^{l+1}[t], \quad \left( \frac{\partial L_E[t]}{\partial \boldsymbol{W}^l} \right)_{\text{OT}} = \widehat{\boldsymbol{a}}^l[t] \frac{\partial L_E[t]}{\partial \boldsymbol{s}^N[t]} \left( \prod_{i=N-1}^{l+1} \frac{\partial \boldsymbol{s}^{i+1}[t]}{\partial \boldsymbol{s}^i[t]} \right) \frac{\partial \boldsymbol{s}^{l+1}[t]}{\partial \boldsymbol{u}^{l+1}[t]}.$$

We show that the gradient of SAF-E is equal to the gradient of OTTT$_O$ by transforming $\boldsymbol{g}_{\widehat{\boldsymbol{U}}}^{l+1}[t]$. First, we calculate $\partial L_E[t]/\partial \widehat{\boldsymbol{a}}^N[t]$. Because $L_E[t]$ does not include any argument up to $t-1$, it holds that

$$\frac{\partial L_E[t]}{\partial \boldsymbol{a}^N[t]} = \frac{\partial L_E[t]}{\partial \boldsymbol{s}^N[t]} \frac{\partial \boldsymbol{s}^N[t]}{\partial \widehat{\boldsymbol{a}}^N[t]} = \frac{\partial L_E[t]}{\partial \boldsymbol{s}^N[t]}.$$

Then, we have

$$\boldsymbol{g}_{\widehat{\boldsymbol{U}}}^{l+1}[t] = \frac{\partial L_E[t]}{\partial \boldsymbol{s}^N[t]} \left( \prod_{i=N-1}^{l+1} \frac{\partial \widehat{\boldsymbol{a}}^{i+1}[t]}{\partial \widehat{\boldsymbol{a}}^i[t]} \right) \frac{\partial \widehat{\boldsymbol{a}}^{l+1}[t]}{\partial \widehat{\boldsymbol{U}}^{l+1}[t]}.$$

Second, because of the forward process of SAF, i.e., (4) and (5), we obtain that

$$\frac{\partial \widehat{\boldsymbol{a}}^{l+1}[t]}{\partial \widehat{\boldsymbol{U}}^{l+1}[t]} = \delta(\widehat{\boldsymbol{U}}^{l+1}[t] - V_{\text{th}}(\lambda \widehat{\boldsymbol{a}}^{l+1}[t-1] + 1)$$

$$= \delta(\boldsymbol{u}^{l+1}[t] - V_{\text{th}}),$$

where $\delta$ represents the delta function. On the other hand, the following equation can be derived from the forward process of OTTT$_O$, i.e., (1):

$$\frac{\partial \boldsymbol{s}^{l+1}[t]}{\partial \boldsymbol{u}^{l+1}[t]} = \delta(\boldsymbol{u}^{l+1}[t] - V_{\text{th}}).$$

Hence,

$$\frac{\partial \widehat{\boldsymbol{a}}^{l+1}[t]}{\partial \widehat{\boldsymbol{U}}^{l+1}[t]} = \frac{\partial \boldsymbol{s}^{l+1}[t]}{\partial \boldsymbol{u}^{l+1}[t]}. \tag{15}$$

Then, we have that

$$\boldsymbol{g}_{\widehat{\boldsymbol{U}}}^{l+1}[t] = \frac{\partial L_E[t]}{\partial \boldsymbol{s}^N[t]} \left( \prod_{i=N-1}^{l+1} \frac{\partial \widehat{\boldsymbol{a}}^{i+1}[t]}{\partial \widehat{\boldsymbol{a}}^i[t]} \right) \frac{\partial \boldsymbol{s}^{l+1}[t]}{\partial \boldsymbol{u}^{l+1}[t]}.$$

From (4), $\widehat{\boldsymbol{a}}^{i+1}[t]$ depends on $\widehat{\boldsymbol{a}}^{i+1}[t-1]$ and $\widehat{\boldsymbol{U}}^i[t]$, while $\widehat{\boldsymbol{a}}^{i+1}[t-1]$ does not depend on $\widehat{\boldsymbol{a}}^{i+1}[t]$, Thus, we calculate that

$$\frac{\partial \widehat{\boldsymbol{a}}^{i+1}[t]}{\partial \widehat{\boldsymbol{a}}^i[t]} = \frac{\partial \widehat{\boldsymbol{a}}^{i+1}[t]}{\partial \widehat{\boldsymbol{U}}^{i+1}[t]} \frac{\partial \widehat{\boldsymbol{U}}^{i+1}[t]}{\partial \widehat{\boldsymbol{a}}^i[t]} + \frac{\partial \widehat{\boldsymbol{a}}^{i+1}[t]}{\partial \widehat{\boldsymbol{a}}^{i+1}[t-1]} \frac{\partial \widehat{\boldsymbol{a}}^{i+1}[t-1]}{\partial \widehat{\boldsymbol{a}}^i[t]} = \frac{\partial \widehat{\boldsymbol{a}}^{i+1}[t]}{\partial \widehat{\boldsymbol{U}}^{i+1}[t]} \boldsymbol{W}^l. \tag{16}$$

Moreover, $\boldsymbol{s}^{i+1}[t]$ depends on $\boldsymbol{u}^{i+1}[t]$, because the forward process of $\text{OTTT}_\text{O}$ is given by (1). Hence,

$$\frac{\partial \boldsymbol{s}^{i+1}[t]}{\partial \boldsymbol{s}^i[t]} = \frac{\partial \boldsymbol{s}^{i+1}[t]}{\partial \boldsymbol{u}^{i+1}[t]} \frac{\partial \boldsymbol{u}^{i+1}[t]}{\partial \boldsymbol{s}^i[t]} = \frac{\partial \boldsymbol{s}^{i+1}[t]}{\partial \boldsymbol{u}^{i+1}[t]} \boldsymbol{W}^l. \tag{17}$$

Combining (15), (16) and (17), we have

$$\frac{\partial \widehat{\boldsymbol{a}}^{i+1}[t]}{\partial \widehat{\boldsymbol{a}}^i[t]} = \frac{\partial \boldsymbol{s}^{i+1}[t]}{\partial \boldsymbol{s}^i[t]}.$$

In the end, we transform $\boldsymbol{g}_{\widehat{\boldsymbol{U}}}^{l+1}[t]$ as follows:

$$\boldsymbol{g}_{\widehat{\boldsymbol{U}}}^{l+1}[t] = \frac{\partial L_E[t]}{\partial \boldsymbol{s}^N[t]} \left( \prod_{i=N-1}^{l+1} \frac{\partial \boldsymbol{s}^{i+1}[t]}{\partial \boldsymbol{s}^i[t]} \right) \frac{\partial \boldsymbol{s}^{l+1}[t]}{\partial \boldsymbol{u}^{l+1}[t]}. \tag{18}$$

This concludes the proof. $\qquad\square$

### A.4    Proof of Theorem 2

**Theorem 2.** *Suppose that $\boldsymbol{m}[t]$ converges when $t \to \infty$. Then, for sufficiently large $T$, the backward processes of SAF-F and Spike Representation are essentially identical, that is, $\frac{\partial L_F}{\partial \boldsymbol{W}^l} = V_\text{th} \left( \frac{\partial L_F}{\partial \boldsymbol{W}^l} \right)_\text{SR}.$*

*Proof.* From assumptions, $\boldsymbol{a}^{l+1}[T] \approx \sigma\big((\boldsymbol{W}^l \boldsymbol{a}^l[T] + \boldsymbol{b}^{l+1})/V_\text{th}\big)$, where $\sigma(x) = \min(\max(0, x), 1)$. Therefore, the followings hold for $i = 1, \ldots, N-1$ :

$$\frac{\partial L_F}{\partial \boldsymbol{W}^l} = \frac{\partial L_F}{\partial \boldsymbol{a}^N[T]} \frac{\partial \boldsymbol{a}^N[T]}{\partial \boldsymbol{W}^l}, \tag{19}$$

$$\frac{\partial \boldsymbol{a}^{i+1}[T]}{\partial \boldsymbol{W}^l} = \frac{\partial \boldsymbol{a}^{i+1}[T]}{\partial \boldsymbol{a}^i[T]} \frac{\partial \boldsymbol{a}^i[T]}{\partial \boldsymbol{W}^l}. \tag{20}$$

By repeatedly substituting (20) for (19), we can calculate $\partial L_F/\partial \boldsymbol{W}^l$ as follows:

$$\frac{\partial L_F}{\partial \boldsymbol{W}^l} = \frac{\partial L_F}{\partial \boldsymbol{a}^N[T]} \left( \prod_{i=N-1}^{l+1} \frac{\partial \boldsymbol{a}^{i+1}[T]}{\partial \boldsymbol{a}^i[T]} \right) \frac{\partial \boldsymbol{a}^{l+1}[T]}{\partial \boldsymbol{W}^l}. \tag{21}$$

This is the gradient of Spike Representaion and denote it by $(\partial L_F/\partial \boldsymbol{W}^l)_\text{SR}$. We will show that $(\partial L_F/\partial \boldsymbol{W}^l)_\text{SR}$ is in proportion to $\partial L_F/\partial \boldsymbol{W}^l = \widehat{\boldsymbol{a}}^l[T] \boldsymbol{g}_{\widehat{\boldsymbol{U}}}^{l+1}[T]$, which is the gradient of SAF-F.

First, let $\Lambda = \sum_{\tau=0}^T \lambda^{T-\tau}$. Then $\boldsymbol{a}^l[T] = \widehat{\boldsymbol{a}}^l[T]/\Lambda$ for any layer $l$. From linearity of differentiation and change of variables, we have followings:

$$\frac{\partial L_F}{\partial \boldsymbol{a}^N[T]} = \frac{\partial L_F}{\Lambda \partial \widehat{\boldsymbol{a}}^N[T]}, \quad \frac{\partial \boldsymbol{a}^{i+1}[T]}{\partial \boldsymbol{a}^i[T]} = \frac{\partial \widehat{\boldsymbol{a}}^{i+1}[T]}{\partial \widehat{\boldsymbol{a}}^i[T]}, \quad \frac{\partial \boldsymbol{a}^{l+1}[T]}{\partial \boldsymbol{W}^l} = \frac{\Lambda \partial \widehat{\boldsymbol{a}}^{l+1}[T]}{\partial \boldsymbol{W}^l}. \tag{22}$$

Substituting (22) into (21), then we obtain

$$\left( \frac{\partial L_F}{\partial \boldsymbol{W}^l} \right)_\text{SR} = \frac{\partial L_F}{\partial \widehat{\boldsymbol{a}}^N[T]} \left( \prod_{i=N-1}^{l+1} \frac{\partial \widehat{\boldsymbol{a}}^{i+1}[T]}{\partial \widehat{\boldsymbol{a}}^i[T]} \right) \frac{\partial \widehat{\boldsymbol{a}}^{l+1}[T]}{\partial \boldsymbol{W}^l}.$$

Second, it follows from $\boldsymbol{a}^l[T] = \widehat{\boldsymbol{a}}^l[T]/\Lambda$ and $\boldsymbol{a}^{l+1}[T] \approx \sigma\big((\boldsymbol{W}^l \boldsymbol{a}^l[T] + \boldsymbol{b}^{l+1})/V_\text{th}\big)$ that

$$\widehat{\boldsymbol{a}}^{l+1}[T] \approx \Lambda \sigma \left( \frac{1}{V_\text{th}} \left( \frac{1}{\Lambda} \boldsymbol{W}^l \widehat{\boldsymbol{a}}^l[T] + \boldsymbol{b}^{l+1} \right) \right).$$

Here, taking care that $\sigma$ is element-wise, we calculate as follows:

$$
\left(\frac{\partial L_F}{\partial \boldsymbol{W}^l}\right)_{\mathrm{SR}} = \frac{\widehat{\boldsymbol{a}}^l[T]}{V_{\mathrm{th}}\Lambda}\left(\frac{\partial L_F}{\partial \widehat{\boldsymbol{a}}^N[T]}\left(\prod_{i=N-1}^{l+1}\frac{\partial \widehat{\boldsymbol{a}}^{i+1}[T]}{\partial \widehat{\boldsymbol{a}}^i[T]}\right)\odot \Lambda\sigma'\left(\frac{1}{V_{\mathrm{th}}}\left(\frac{1}{\Lambda}\boldsymbol{W}^l\widehat{\boldsymbol{a}}^l[T]+\boldsymbol{b}^{l+1}\right)^{\top}\right)\right)
$$

$$
= \frac{\widehat{\boldsymbol{a}}^l[T]}{V_{\mathrm{th}}}\left(\frac{\partial L_F}{\partial \widehat{\boldsymbol{a}}^N[T]}\left(\prod_{i=N-1}^{l+1}\frac{\partial \widehat{\boldsymbol{a}}^{i+1}[T]}{\partial \widehat{\boldsymbol{a}}^i[T]}\right)\odot \boldsymbol{d}^{l+1}[T]^{\top}\right),
$$

where we set $\boldsymbol{d}^{l+1}[T] = \sigma'\left((\boldsymbol{W}^l\widehat{\boldsymbol{a}}^l[T]/\Lambda+\boldsymbol{b}^{l+1})/V_{\mathrm{th}}\right)$, and $\odot$ is the element-wise product. Now we assume that

$$
\frac{\partial \boldsymbol{s}^{l+1}[T]}{\partial \boldsymbol{u}^{l+1}[T]} = \mathrm{diag}(\boldsymbol{d}^{l+1}[T]),
$$

for any $l = 0,\ldots,N-1$, where $\mathrm{diag}(\boldsymbol{d}^{l+1}[T])$ is a diagonal matrix constructed from $\boldsymbol{d}^{l+1}[T]$. The reason why this assumption is valid discussed in Xiao et al. (2022). Then, we obtain from (15) that

$$
\left(\frac{\partial L_F}{\partial \boldsymbol{W}^l}\right)_{\mathrm{SR}} = \frac{\widehat{\boldsymbol{a}}^l[T]}{V_{\mathrm{th}}}\frac{\partial L_F}{\partial \widehat{\boldsymbol{a}}^N[T]}\left(\prod_{i=N-1}^{l+1}\frac{\partial \widehat{\boldsymbol{a}}^{i+1}[T]}{\partial \widehat{\boldsymbol{a}}^i[T]}\right)\frac{\partial \widehat{\boldsymbol{a}}^{l+1}[T]}{\partial \widehat{\boldsymbol{U}}^{l+1}[T]} = \frac{1}{V_{\mathrm{th}}}\widehat{\boldsymbol{a}}^l[T]\,\boldsymbol{g}_{\widehat{U}}^{l+1}[T].
$$

$\square$

# B  FEEDFORWARD AND FEEDBACK CONNECTION

## B.1  FEEDFORWARD CONNECTION

The forward process of the $(q+1)$-th layer of the SNN with a feedforward connection from the $p$-th layer to the $(q+1)$-th layer (where $q \geq p$) with weight $\boldsymbol{W}_f$ is as follows:

$$
\begin{cases}
\boldsymbol{u}^{q+1}[t] = \lambda(\boldsymbol{u}^{q+1}[t-1] - V_{\mathrm{th}}\,\boldsymbol{s}^{q+1}[t-1]) + \boldsymbol{W}^q\boldsymbol{s}^q[t] + \boldsymbol{b}^{q+1} + \boldsymbol{W}_f\boldsymbol{s}^p[t], \\
\boldsymbol{s}^{q+1}[t] = H(\boldsymbol{u}^{q+1}[t] - V_{\mathrm{th}}).
\end{cases}
$$

Note that the layers other than the $(q+1)$-th layer are the same as in (1). Meanwhile, the forward process of the $(q+1)$-th layer of the SAF is as follows:

$$
\begin{cases}
\widehat{\boldsymbol{U}}^{q+1}[t] = \boldsymbol{W}^q\widehat{\boldsymbol{a}}^q[t] + \boldsymbol{b}^{q+1}(\sum_{\tau=0}^{t-1}\lambda^{\tau}) + \lambda^t\widehat{\boldsymbol{U}}^{q+1}[0] + \boldsymbol{W}_f\,\widehat{\boldsymbol{a}}^p[t], \\
\widehat{\boldsymbol{a}}^{q+1}[t] = \widehat{\boldsymbol{a}}^{q+1}[t-1] + H(\widehat{\boldsymbol{U}}^{q+1} - V_{\mathrm{th}}(\lambda\widehat{\boldsymbol{a}}^{q+1}[t-1] + 1)).
\end{cases}
$$

As with LIF neurons, the layers other than the $(q+1)$-th layer are the same as in (4). The forward processes of SAF and LIF with feedforward connection are mutually convertible.
Regarding the backward processes, the gradients for parameters other than $\boldsymbol{W}_f$ are the same as when there is no feedforward connection. The derivative with respect to $\boldsymbol{W}_f$ is caluclated as well as in Sec. 4.3. Therefore, $\partial L_E[t]/\partial \boldsymbol{W}_b = \widehat{\boldsymbol{a}}^p[t]\,\boldsymbol{g}_{\widehat{U}}^{q+1}[t]$ for SAF-E and $\partial L_F/\partial \boldsymbol{W}_b = \widehat{\boldsymbol{a}}^p[T]\,\boldsymbol{g}_{\widehat{U}}^{q+1}[T]$ for SAF-F. Thus, it can be confirmed that SAF-E is equivalent to OTTT$_{\mathrm{O}}$. Also, the gradient descent directions in SAF-F and Spike Representation are close.

## B.2  EQUIVALENCE WITH OTTT$_{\mathrm{O}}$ AND PROXIMITY TO SPIKE REPRESENTATION

In this subsection, we consider only the case of feedback connections, but we can check that the followings hold for feedforward connections as well.

**Equivalence with OTTT$_{\mathrm{O}}$**

We show the equivalence of SAF-E and OTTT$_{\mathrm{O}}$ with a feedback connection. We assume that the SNN has a feedback connection from the $p$-th layer to the $(q+1)$-th layer (where $q < p$) with weight $\boldsymbol{W}_b$. As

mentioned in Sec. 4.3, the gradients for parameters other than $\boldsymbol{W}_b$ are equal to the gradients when there is no feedback connection. From Theorem 1, these gradients are same as $\text{OTTT}_\text{O}$. Therefore, we only need to check $\partial L_E[t]/\partial \boldsymbol{W}_b = (\partial L_E[t]/\partial \boldsymbol{W}_b)_\text{OT}$. In fact, the gradient of $\text{OTTT}_\text{O}$ calculated (see Xiao et al. (2022)) as

$$\left(\frac{\partial L_E[t]}{\partial \boldsymbol{W}_b}\right)_\text{OT} = \widehat{\boldsymbol{a}}^p[t-1]\frac{\partial L[t]}{\partial \boldsymbol{s}^N[t]}\left(\prod_{i=N-1}^{q+1}\frac{\partial \boldsymbol{s}^{i+1}[t]}{\partial \boldsymbol{s}^i[t]}\right)\frac{\partial \boldsymbol{s}^{q+1}[t]}{\partial \boldsymbol{u}^{q+1}[t]},$$

and (18) holds, then we have

$$\left(\frac{\partial L_E[t]}{\partial \boldsymbol{W}_b}\right)_\text{OT} = \widehat{\boldsymbol{a}}^p[t-1]\,\boldsymbol{g}_{\widehat{U}}^{q+1}[t] = \frac{\partial L_E[t]}{\partial \boldsymbol{W}_b}.$$

Hence, SAF-E is equivalent to $\text{OTTT}_\text{O}$ even if there is a feedback connection in SNN.

**Proximity to Spike Representation**

Assume that the SNN is the same as in Sec. 4.3. Then, the same assertion as Theorem 2 does not hold, that is, SAF-F is not equivalent to Spike Representation in general. However, we can show that the gradient descent directions in SAF-F and Spike Representation are close.

**Theorem 3.** *Suppose that $\boldsymbol{m}[t]$ converges when $t \to \infty$. Then, for sufficiently large $T$, the backward processes of SAF-F and Spike Representation with a feedback connection are close, that is, $\left\langle\frac{\partial L_F}{\partial \boldsymbol{\theta}}, \left(\frac{\partial L_F}{\partial \boldsymbol{\theta}}\right)_\text{SR}\right\rangle > 0$ for all parameters $\boldsymbol{\theta}$.*

*Proof.* From the assumption, the firing rates of each layer converge to these equilibrium points: $(\boldsymbol{a}^{l+1})^* = f_{l+1}((\boldsymbol{a}^l)^*)$ $(l \neq q)$, $(\boldsymbol{a}^{q+1})^* = f_{q+1}(f_p \circ \cdots \circ f_{q+2}((\boldsymbol{a}^{q+1})^*)), (\boldsymbol{a}^q)^* = f_{q+1}((\boldsymbol{a}^p)^*, (\boldsymbol{a}^q)^*)$, where $f_{l+1}((\boldsymbol{a}^l)^*) = \sigma\left((\boldsymbol{W}^l(\boldsymbol{a}^l)^* + \boldsymbol{b}^{l+1})/V_\text{th}\right)$, and $f_{l+1}((\boldsymbol{a}^l)^*) = \sigma\left((\boldsymbol{W}^l(\boldsymbol{a}^l)^* + \boldsymbol{b}^{l+1})/V_\text{th}\right)$. Since $T \gg 1$, $\partial L_F/\partial \boldsymbol{\theta}$ can be calculated using the spike representation as follows (see Xiao et al. (2021),Xiao et al. (2022)):

$$\left(\frac{\partial L_F}{\partial \boldsymbol{\theta}}\right)_\text{SR} = \frac{\partial L_F}{\partial \boldsymbol{a}^{l+1}[T]}\left(I - \frac{\partial f_{l+1}}{\partial \boldsymbol{a}^l[T]}\right)^{-1}\frac{\partial f_{l+1}(\boldsymbol{a}^l[T])}{\partial \boldsymbol{\theta}}, \tag{23}$$

where $\partial f_{l+1}/\partial \boldsymbol{a}^l[T]$ denotes Jacobian matrix and $I$ denotes identity matrix. Here, we regard $(I - \partial f_{l+1}/\partial \boldsymbol{a}^l[T])^{-1}$ of (23) as an identity matrix, and denote it by

$$\left(\widetilde{\frac{\partial L_F}{\partial \boldsymbol{\theta}}}\right)_\text{SR} = \frac{\partial L_F}{\partial \boldsymbol{a}^{l+1}[T]}\frac{\partial f_{l+1}(\boldsymbol{a}^l[T])}{\partial \boldsymbol{\theta}}. \tag{24}$$

Then, it is proved in Fung et al. (2022) and Geng et al. (2021) that

$$\left\langle\left(\widetilde{\frac{\partial L_F}{\partial \boldsymbol{\theta}}}\right)_\text{SR}, \left(\frac{\partial L_F}{\partial \boldsymbol{\theta}}\right)_{SR}\right\rangle > 0,$$

where $\langle\cdot,\cdot\rangle$ denotes inner product. If $\boldsymbol{\theta}$ is not $\boldsymbol{W}_b$, right-hand side of (24) is equal to right-hand side of (21), then from Theorem 2,

$$\left(\widetilde{\frac{\partial L_F}{\partial \boldsymbol{\theta}}}\right)_\text{SR} = \frac{1}{V_\text{th}}\frac{\partial L_F}{\partial \boldsymbol{\theta}}.$$

Next, we consider $\boldsymbol{\theta}$ as $\boldsymbol{W}_b$. $\boldsymbol{a}^{q+1}[T]$ can be approximated by $\sigma\left((\boldsymbol{W}^q\boldsymbol{a}^q[T] + \boldsymbol{b}^{q+1} + \boldsymbol{W}_b\,\boldsymbol{a}^p[T])/V_\text{th}\right)$ and $\boldsymbol{a}^p[T] \approx \boldsymbol{a}^p[T-1]$ because $T$ is large. Therefore, $\boldsymbol{a}^{q+1}[T] \approx \sigma\left((\boldsymbol{W}^q\boldsymbol{a}^q[T] + \boldsymbol{b}^{q+1} + \boldsymbol{W}_b\,\boldsymbol{a}^p[T-1])/V_\text{th}\right)$. Setting $\boldsymbol{d}_b^{l+1}[T] = \sigma'\left((\boldsymbol{W}^l\boldsymbol{a}^l[T] + \boldsymbol{b}^{l+1} + \boldsymbol{W}_b\,\boldsymbol{a}^k[T-1])/V_\text{th}\right)$ and calculating similar to Sec. A.4, we derive

that

$$\left(\widetilde{\frac{\partial L_F}{\partial \boldsymbol{W}_b}}\right)_{\mathrm{SR}} = \frac{\widehat{\boldsymbol{a}}^k[T-1]}{V_{\mathrm{th}}} \left(\frac{\partial L_F}{\partial \widehat{\boldsymbol{a}}^N[T]} \left(\prod_{i=N-1}^{l+1} \frac{\partial \widehat{\boldsymbol{a}}^{i+1}[T]}{\partial \widehat{\boldsymbol{a}}^i[T]}\right) \odot \boldsymbol{d}_b^{l+1}[T]\right)$$

$$= \frac{1}{V_{\mathrm{th}}} \widehat{\boldsymbol{a}}^p[T-1] \, \boldsymbol{g}_{\widehat{U}}^{q+1}[T]$$

$$= \frac{1}{V_{\mathrm{th}}} \left(\frac{\partial L_F}{\partial \boldsymbol{W}_b}\right).$$

In the end, take the inner product between their gradient we obtain that

$$\left\langle \left(\frac{\partial L_F}{\partial \boldsymbol{\theta}}\right), \left(\frac{\partial L_F}{\partial \boldsymbol{\theta}}\right)_{SR} \right\rangle = V_{\mathrm{th}} \left\langle \left(\widetilde{\frac{\partial L_F}{\partial \boldsymbol{\theta}}}\right)_{\mathrm{SR}}, \left(\frac{\partial L_F}{\partial \boldsymbol{\theta}}\right)_{SR} \right\rangle > 0.$$

$\square$

## C  Implementation detail

In the experiments of our study, we used the VGG network as follows:

64C3-128C3-AP2-256C3-256C3-AP2-512C3-512C3-AP2-512C3-512C3-GAP-FC,

where $x$C$y$ represents the convolutional layer with $x$-output channels and $y$-stride, AP$x$ represents the average pooling with the kernel size 2, GAP represents the global average pooling, and FC represents the fully connected layer.

As for the time step, we set it to 6 if there is no mention. We used the stochastic gradient descent (SGD) as the optimizer with the batch size, epoch, initial learning rate for the cosine annealing, and momentum at 128, 300, 0.1, and 0.9. As for the loss function, we applied the combination of the cross-entropy (CE) and the mean-squared error (MSE) losses, i.e., $L = (1 - \alpha)\mathrm{CE} + \alpha\mathrm{MSE}$ (where $\alpha = 0.05$). In addition, we set the leaky term as $\lambda = 0.5$ and the threshold as $V_{\mathrm{th}} = 1$. Note that all settings, including the above, were the same as Xiao et al. (2022).

## D  Pseudocode

We show the pseudo-code of SAF-E and SAF-F in Algorithm 1 to better understand our methods.

## E  Experimental Analysis of SAF-E with Feedback Connection

In this section, we compare the performance of SAF-E with OTTT$_{\mathrm{O}}$ when both networks have a feedback connection from the first layer to the $N$-th layer. The setup was the same as in the experiments of Xiao et al. (2022).

Table 3 shows the accuracy and total firing rate when we set $T = 6$. As shown in this table, SAF-E and OTTT$_{\mathrm{O}}$ with feedback connection are almost identical. Also, the change due to inference with SNNs consisting of LIF neurons is also almost negligible. Figure 6 shows the accuracy, loss curve, and the firing rates of each layer. From these results, we confirmed experimentally that SAF-E and OTTT$_{\mathrm{O}}$ with feedback connection are equivalent, as mentioned in Sec. B.2.

---

**Algorithm 1** One iteration of SAF training.

---

**Require:** Network parameters $\{\boldsymbol{W}^l\}$, $\{\boldsymbol{b}^{l+1}\}$; Time steps $T$; Number of layers $N$; Other hyperparameters; Input dataset
**Ensure:** Trained network parameters $\{\boldsymbol{W}^l\}$, $\{\boldsymbol{b}^{l+1}\}$
1: **for** $t = 1, 2, \ldots, T$ **do**
2:     % Forward
3:     **for** $l = 1, 2, \ldots, N$ **do**
4:         Update the (weighted) potential accumulation $\widehat{\boldsymbol{U}}^l[t]$ and spike accumulation $\widehat{\boldsymbol{a}}^l[t]$ using (4).
5:     **end for**
6:     % Backward
7:     **for** $l = N, N-1, \ldots, 1$ **do**
8:         **if** training option is SAF-E **then**
9:             Update parameters with $\partial L_E[t]/\partial \boldsymbol{W}^l = \widehat{\boldsymbol{a}}^l[t]\, \boldsymbol{g}_{\widehat{U}}^{l+1}[t]$ based on the gradient-based optimizer.
10:         **else if** training option is SAF-F **and** $t = T$ **then**
11:             Update parameters with $\partial L_F/\partial \boldsymbol{W}^l = \widehat{\boldsymbol{a}}^l[T]\, \boldsymbol{g}_{\widehat{U}}^{l+1}[T]$ based on the gradient-based optimizer.
12:         **end if**
13:     **end for**
14: **end for**

---

Table 3: Comparison of SAF-E and OTTT$_\text{O}$ with feedback connection. The values in parentheses are the changes in firing rate and accuracy due to inference by the SNN composed of LIF neurons.

| Method | Firing rate [%] | Accuracy [%] |
|---|---|---|
| OTTT$_\text{O}$ with feedback | 14.74$\pm$0.34 | 93.23$\pm$0.28 |
| SAF-E with feedback (ours) | 14.32$\pm$0.12 ($1.78 \times 10^{-6}$) | 93.20$\pm$0.18 ($6.67 \times 10^{-5}$) |

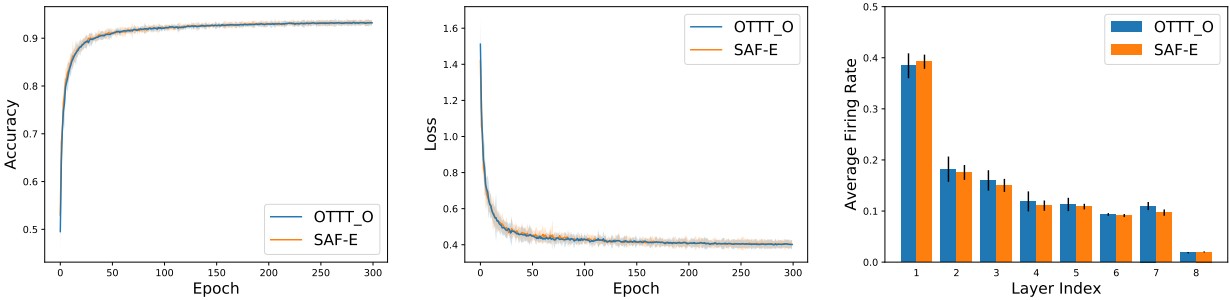

Figure 6: Accuracy and loss curves, and firing rates of each layer of SAF-E and OTTT$_\text{O}$ with feedback connection.

