# OpenReview forum: "Spike Accumulation Forwarding for Effective Training of Spiking Neural Networks"
_ICLR.cc/2024/Conference — Submitted to ICLR 2024_

### Official Review · Reviewer_q39D · 2023-10-24

**Soundness:** 3 good
**Presentation:** 2 fair
**Contribution:** 3 good
**Rating:** 5
**Confidence:** 5

**Summary:**

This article introduces a new method for training Spiking Neural Networks (SNNs) known as Spike Accumulation Forwarding (SAF). The authors conduct theoretical analysis and experimental comparisons to establish the equivalence of SAF with other methods, such as OTTT, while evaluating its performance and efficiency. Experimental data demonstrates that SAF can significantly reduce training time and memory usage with little to no loss in accuracy. Additionally, it is shown that SAF-trained parameters can be used for inference in SNNs composed of LIF neurons.

**Strengths:**

The article introduces an innovative method, SAF, for training SNNs, and its effectiveness is demonstrated through both theoretical analysis and experiments.

**Weaknesses:**

1. The relatively simple dataset used in the article (CIFAR-10) may limit the understanding of the method's applicability to more complex datasets.
2. The Figure 2 in the experimental section is relatively blurry.
3. The memory compression in Table 1 is also somewhat limited. Are there better results available to demonstrate the effectiveness of the method?
4. The experimental section lacks specific details about the SNN model, and it would be beneficial to test the method on different models.

**Questions:**

Do the authors have plans to apply the SAF method to other types of neural networks or larger datasets? Are there plans for more experiments to validate SAF's performance in different scenarios?

---

> ### Author Response · Authors · 2023-11-17
> **Response to Reviewer q39D**
>
> Thank you for your review comments. We reply to the weak points and answer your questions. For clarity, we have addressed each of your comments individually in the following responses:
>
> **---1.The relatively simple dataset used in the article (CIFAR-10) may limit the understanding of the method's applicability to more complex datasets.
> ---4.The experimental section would be beneficial to test the method on different models.
> ---Do the authors have plans to apply the SAF method to other types of neural networks or larger datasets? Are there plans for more experiments to validate SAF's performance in different scenarios?**
> In Section 5, our main objective is to ensure that there are no inconsistencies between our theory and experiment. Our theory suggests that SAF-E is equivalent to OTTT$_ {\\rm O}$, and SAF-F is equivalent to spike representation. These equivalences can be confirmed through the current experiments. Additionally, it is worth noting that OTTT and Spike Representation, as mentioned in Xiao et al. [2022], can be applied to more complex datasets. Given that our methods are in full theoretical agreement with these approaches, this fact clearly supports the scalability of our method. Therefore, it is evident that scalability can be demonstrated without the large and complex experiments. We are willing to add it to our paper if necessary.
>
> **---2.The Figure 2 in the experimental section is relatively blurry.**
> The blurriness in Figure 2 is indeed a result of the overlap of lines, which is expected because SAF-E and OTTT$_ {\\rm O}$ are theoretically equivalent. Conversely, in Figure 4, the non-overlapping lines representing SAF-F and OTTT$_ {\\rm A}$ indicate that they are not equivalent. This result also aligns with our theoretical predictions.
>
>
> **---3.The memory compression in Table 1 is also somewhat limited. Are there better results available to demonstrate the effectiveness of the method?**
> Compared to OTTT, SAF achieved a notable reduction of 30\% in memory usage. Similarly, the training time was also reduced by the same percentage, at least. These results are not only significant but also demonstrating substantial differences between the two methods.
>
> **---4.The experimental section lacks specific details about the SNN model,**
> The details of the model used and the training setup can be found in Appendix C, and the source code is available in the Supplementary material. It is important to note that all setups were consistent with the experiments conducted by Xiao et al. [2022].

---

> > ### Comment · Reviewer_q39D · 2023-11-23
> > **Thanks for the clarification**
> >
> > I still have concerns about large-scale datasets. I am wondering if the proposed method applies to more complex datasets.

---

### Official Review · Reviewer_owpx · 2023-10-25

**Soundness:** 2 fair
**Presentation:** 2 fair
**Contribution:** 2 fair
**Rating:** 3
**Confidence:** 4

**Summary:**

Spike accumulation forwarding (SAF) is a method that reduces memory requirements and training time for SNNs by accumulating spikes over multiple time steps. This approach is consistent with Spike Representation and online training through time methods. The authors tested SAF's effectiveness on the CIFAR-10 dataset.

**Strengths:**

The authors tried to solve an important challenge with training SNNs, namely the need for large amounts of memory and long training times. And the only small improvement I see is in memory and training times.

**Weaknesses:**

The authors didn’t compare SAF to BPTT and other successful training methods, requiring a comprehensive understanding.

They conducted experiments on a single set of hypermeters, which lacks further exploration of the robustness of the methods. Understanding the effects of various hyperparameters on SAF's performance and how to tune them for various applications would be improved by a more thorough analysis.

The authors have not provided a detailed analysis of the computational complexity of SAF. The authors have empirically demonstrated that SAF reduces memory needs and training time; however, they haven't done a thorough investigation of the computational complexity of the approach. The experiment is currently based on GPU architecture. It would be easier to comprehend the computational resources needed to execute SAF and its potential for real-world applications with a more thorough analysis.

The theoretical results only show that the proposed method has the same capabilities as OTTT and SpikeRepresentation, and no new guarantee of training is presented there. Hence, the contribution is reduced to find an efficient forward method that fits the training models of OTTT and SpikeRepresentation.

**Questions:**

Can you provide detailed explanations of how much computational overhead is needed when training? What kind of computation is included (float-point computation or spike computation)? I am interested in this because the training, unlike other online training, is based on float-point representation.

Regarding the robustness of the training algorithm, can you provide some results that have different settings of $\lambda$ and $T$.

Also, the current dataset setting is lacking. Only CIFAR-10 experiments are conducted. What about CIFAR-100 and ImageNet?

Overall, I am more inclined to reject this paper.

---

> ### Author Response · Authors · 2023-11-17
> **Response to Reviewer owpx 1/2**
>
> Thank you for your review comments. We reply to the weak points and answer your questions. For clarity, we have addressed each of your comments individually in the following responses:
>
> **---The authors didn’t compare SAF to BPTT and other successful training methods, requiring a comprehensive understanding.**
> The primary objective of this paper is indeed to establish the theoretical consistency between SAF-E and OTTT$_{\rm O}$, as well as SAF-F and spike representation. Therefore, comparisons with other methods are beyond the scope of the paper's claims. It is worth noting that previous studies have already demonstrated the superior performance of OTTT over BPTT and other methods.
>
> **---They conducted experiments on a single set of hypermeters, which lacks further exploration of the robustness of the methods. Understanding the effects of various hyperparameters on SAF's performance and how to tune them for various applications would be improved by a more thorough analysis.
> ---Regarding the robustness of the training algorithm, can you provide some results that have different settings of $\mathbf{\lambda}$ and $\mathbf{T}$.**
> As mentioned earlier, the paper indeed provides a theoretical guarantee of consistency between SAF-E and OTTT$_{\rm O}$, as well as SAF-F and spike representations. Additionally, it is worth noting that the hyperparameters of SAF are exactly the same as the hyperparameters of OTTT. This implies that conducting a hyperparameter search will yield identical accuracies to OTTT.  However, it is important to note that time and memory usage are not explicitly addressed in the theoretical analysis. To provide a more comprehensive understanding of the practical implications, Figures 3 (A, B) and 5 (A, B) present the training time and memory usage when varying $T$, as it has the most significant impact on these indicators. This allows for a more complete evaluation of the performance of the methods beyond accuracy alone.
>
> **---Also, the current dataset setting is lacking. Only CIFAR-10 experiments are conducted. What about CIFAR-100 and ImageNet?**
> In Section 5, our main objective is to ensure that there are no inconsistencies between our theory and experiment. Our theory suggests that SAF-E is equivalent to OTTT$_{\rm O}$, and SAF-F is equivalent to spike representation. These equivalences can be confirmed through the current experiments. Additionally, it is worth noting that OTTT and Spike Representation, as mentioned in Xiao et al. [2022], can be applied to more complex datasets. Given that our methods are in full theoretical agreement with these approaches, this fact clearly supports the scalability of our method. Therefore, it is evident that scalability can be demonstrated without the large and complex experiments. We are willing to add it to our paper if necessary.

---

> ### Author Response · Authors · 2023-11-17
> **Respose to Reviewer owpx 2/2**
>
> **---The authors have not provided a detailed analysis of the computational complexity of SAF. The authors have empirically demonstrated that SAF reduces memory needs and training time; however, they haven't done a thorough investigation of the computational complexity of the approach. The experiment is currently based on GPU architecture. It would be easier to comprehend the computational resources needed to execute SAF and its potential for real-world applications with a more thorough analysis.
> ---Can you provide detailed explanations of how much computational overhead is needed when training? What kind of computation is included (float-point computation or spike computation)? I am interested in this because the training, unlike other online training, is based on float-point representation.**
>
> We compare between the computational complexity (FLOPs) of OTTT and SAF during the table below. $c$, $k$, $w$, and $h$ represent the channel size, kernel size, and the width and height of the feature map. Also, $d_{\rm in}$ and $d_{\rm out}$ represent input and output sizes when fully connected. By using SAF, we can halve the computation of weights.Moreover, SAF only requires the retention of spike accumulation (float), while OTTT needs to retain membrane potentials (float), spike trains (spike), and spike accumulation (float). In other words, SAF only uses floating-point computation during training and spike computation during inference.
> We will add this table to Appendix if necessary.
> |                        | **OTTT$_ {\\rm O}$/OTTT$_ {\\rm A}$** | **SAF-F/SAF-E (ours)** |
> |------------------------|----------------------------------|------------------------|
> | FLOPs (Conv2D)         | $2 c^2 k^2 wh$                   | $c^2 k^2 wh$           |
> | FLOPs (FC)             | $2 d_ {\\rm in} d_ {\\rm out}$       | $d_{\\rm in} d_ {\\rm out}$|
> |------------------------|----------------------------------|------------------------|
> | spike accumulation $\\widehat{\\boldsymbol {a}} ^ l[t]$ | $\\checkmark$ | $\\checkmark $|
> | spike $\\boldsymbol {s} ^ l[t]$    | $\\checkmark$ | -                      |
> | Membrane potential $\widehat{\\boldsymbol {U}}^{l+1}[t]$ |$ \\checkmark$ | -   |
>
> **---The theoretical results only show that the proposed method has the same capabilities as OTTT and SpikeRepresentation, and no new guarantee of training is presented there. Hence, the contribution is reduced to find an efficient forward method that fits the training models of OTTT and SpikeRepresentation.**
> SAF offers memory reduction by eliminating the need to calculate membrane potentials during training, as well as halving the time required for weight calculation during forward propagation. In our experiments, SAF achieved a significant reduction of 30\% in memory usage compared to OTTT. Similarly, the training time was also reduced by the same percentage, at least. These results demonstrate substantial differences between the two methods and highlight the significance of SAF's contributions. Moreover, we have established the theoretical consistency of SAF with OTTT and spike representation for both of forward and backward. Based on these contributions, we believe that our paper is suitable for acceptance by ICLR.

---

### Official Review · Reviewer_J1Az · 2023-10-31

**Soundness:** 3 good
**Presentation:** 2 fair
**Contribution:** 2 fair
**Rating:** 3
**Confidence:** 4

**Summary:**

This paper improves the SNN representation of OTTT method. It simplifies the formulas used in SNN representation forward propagation, thereby reducing the memory and time required in the training phase. The proposed approach in this paper offers insights into SNN training. However, some weaknesses and limitations still remain within the content of this paper.

**Strengths:**

This article presents improvements to OTTT by further simplifying the forward propagation equations. Additionally, the authors demonstrate the consistency between the gradients of their SAF method and OTTT. They provide experimental evidence that their method outperforms OTTT in certain scenarios.

**Weaknesses:**

Weakness
1. There is numerous equations in this paper, and many parts lack detailed reasoning, making it challenging for readers to follow along. For instance, in the section Spike Representation, the intermediate steps of most equations are omitted. And $\mathbf{x}$ lacks explanation.
2. What is the purpose of section 2.4.3. The network structure that included in this paper and OTTT do not have feedback connection.
3. Merely validating the consistency between theoretical reasoning and the experimental results on CIFAR10 is insufficient for the author's purpose. It is evident that the author's approach, despite claiming consistency in the computation of reverse gradients with OTTT, yields different training outcomes (accuracy and fire rate). Moreover, based on the results presented by the author, it is clear that the SNN trained using the SAF method outperforms the OTTT method. Since the OTTT method involves more complex datasets, the author needs to demonstrate the superiority of SAF on those datasets as well.
4. The author mentions that Equation 2 holds true strictly when the network time step T tends to infinity. However, in practice, the author uses relatively small values of T, which introduces errors between the actual spike accumulation of the SNN and the weighted spike accumulation. These errors arise due to the uneven distribution of spikes[1]. Could the author provide insights on the differences between the training results (using representation) and the actual LIF forward results to address this issue?

[1] Bu T, Fang W, Ding J, et al. Optimal ANN-SNN Conversion for High-accuracy and Ultra-low-latency Spiking Neural Networks[C]//International Conference on Learning Representations. 2021.

**Questions:**

See the weakness.

---

> ### Author Response · Authors · 2023-11-17
> **Response to Reviewer J1Az**
>
> Thank you for your review comments. We reply to the weak points and answer your questions. For clarity, we have addressed each of your comments individually in the following responses:
>
> **---1.There is numerous equations in this paper, and many parts lack detailed reasoning, making it challenging for readers to follow along. For instance, in the section Spike Representation, the intermediate steps of most equations are omitted. And $x$ lacks explanation.**
> Due to page limitations, it is only possible to include some reasoning in the main text. The detailed reasoning of Section 4 is provided in Appendix A. Also, $x$ is the value of the input data. In the case of CIFAR-10, it is the value of each pixel in the image. This explanation has been added to the main text.
>
> **---2.What is the purpose of section 2.4.3. The network structure that included in this paper and OTTT do not have feedback connection.**
> The case with feedback is also mentioned in the paper of OTTT. Therefore, we considered SAF with feedback in Section 4.3, and experimental results were stated in Appendix E.
>
> **---3.Merely validating the consistency between theoretical reasoning and the experimental results on CIFAR10 is insufficient for the author's purpose. It is evident that the author's approach, despite claiming consistency in the computation of reverse gradients with OTTT, yields different training outcomes (accuracy and fire rate). Moreover, based on the results presented by the author, it is clear that the SNN trained using the SAF method outperforms the OTTT method. Since the OTTT method involves more complex datasets, the author needs to demonstrate the superiority of SAF on those datasets as well.**
> It is indeed consistent with SAF-F for spike representation, but not with OTTT${_ \\rm A}$. Section 5.2 confirms this distinction, highlighting the differences between the two methods. Consequently, the firing rates are expected to differ, which is correct. On the other hand, Section 5.1 describes the results of an experiment that confirms the equivalence between SAF-E and OTTT${_ \\rm O}$. The figure presented in this section demonstrates the same firing rates. Thanks to your review, we have added explanations to the figure captions in Section 5, improving readability.
>
> **---4.The author mentions that Equation 2 holds true strictly when the network time step T tends to infinity. However, in practice, the author uses relatively small values of T, which introduces errors between the actual spike accumulation of the SNN and the weighted spike accumulation. These errors arise due to the uneven distribution of spikes[1]. Could the author provide insights on the differences between the training results (using representation) and the actual LIF forward results to address this issue?**
> There are two reasons why we set $T=6$. First, OTTT's original paper (Xiao et al. [2022]) sets $T=6$. Second, in the case of CIFAR-10, the original paper on spike representation (Meng et al. [2022]) compares the four cases of $T=20,15,10,5$. The results show that the accuracy drop was less than 1\%, and SOTA was achieved at $T=5$. Since SAF-F and OTTT$_{\rm A}$ are different at $T=6$, it is unnecessary to increase $T$ to obtain similar results.

---

> > ### Comment · Reviewer_J1Az · 2023-11-20
> > **Response to the rebuttal**
> >
> > Thank you for the further explanation. After re-check the content, it seemed that the current method is more or less a simplification rather than an improvement over OTTT. Since OTTT itself is not a widely adopted training framework for SNNs, the impact of the current work is questionable.
> >
> > Also, the reason for asking for results on more complex datasets is that we cannot assume scalability for sure.
> >
> > The explanation for choosing T = 6 is not satisfactory. I understand why they choose T = 6 but it lacks approximation reasoning why choosing T = 6 is a reasonable setup as such a small T may introduce very significant bias.

---

### Official Review · Reviewer_WuBX · 2023-11-01

**Soundness:** 3 good
**Presentation:** 2 fair
**Contribution:** 4 excellent
**Rating:** 5
**Confidence:** 3

**Summary:**

The authors present “Spike Accumulation Forwarding (SAF)” for training spiking neural networks, an approach build on Online Training Through Time (OTTT) to utilize spike representations in form of accumulations through both the forward and the backward pass. This approach helps reduce the memory footprint of SNNs, since the membrane potential of the previous time step does not need to be tracked. In an extensive theoretical analysis, the authors prove the feasibility of their approach as well as its equivalence with the LIF neuron and the consistency with the existing approach OTTT. The theoretical findings are backed by a few pracitical examples.

**Strengths:**

- The approach is novel and useful, as memory footprint in SNN training is a major issue not only in the backward but also in the forward paths.
- The rigorous theoretical analysis proves the authors claims, and the brief experimental evaluation seems to confirm it.

**Weaknesses:**

- The paper is in parts poorly written and very hard to follow. It would benefit from substantial language editing. Also, there are some Sentences and sections, e.g. 3rd paragraph of the introduction, whose meaning I do not understand at all.
- The experimental evaluation is kept rather short. While it does seem to confirm the theoretical findings, there is no clear description of the experiments (e.g. the utilized model architecture and training setup), and no source code, which prohibits reproduction of the results.
- The prior work and related concepts upon which the contribution of the authors build is not explained very well. Half of the assumptions and derivations are to be found in other papers, making it almost impossible to fully grasp the paper as a stand alone. It appears to be follow-up work by the authors of previous work, hence the authors don’t seem to find it necessary to fully explain the background concepts

**Questions:**

- It is not clear to me why the approach works for small time steps. As stated in section 3.2 Spike representation assumes a rather large latency (T -> inf) to work. Does that not also apply to SAF? In Fig. 5 you show that firing rates for OTTT and SAF are NOT Identical. Then why do your experiments assume T=6? Where does the averaged spike representation come from in practice?
- In Section 3, Spike Representation, what is x for the weighted average input? Not mentioned before
- What model architecture and training setup was used for the experiments. As it, they are not reproducible by others.

---

> ### Author Response · Authors · 2023-11-17
> **Response to Reviewer WuBX**
>
> Thank you for your review comments. We reply to the weak points and answer your questions. For clarity, we have addressed each of your comments individually in the following responses:
>
> **---The paper is in parts poorly written and very hard to follow. It would benefit from substantial language editing. Also, there are some Sentences and sections, e.g. 3rd paragraph of the introduction, whose meaning I do not understand at all.**
> We apologize if there was any confusion. However, we have had native English speakers proofread and confirm that there are no problems.
>
> **---The experimental evaluation is kept rather short. While it does seem to confirm the theoretical findings, there is no clear description of the experiments (e.g. the utilized model architecture and training setup), and no source code, which prohibits reproduction of the results.
> ---What model architecture and training setup was used for the experiments. As it, they are not reproducible by others.**
> The details of the model used and the training setup can be found in Appendix C, and the source code is available in the Supplementary material. It is important to note that all setups were consistent with the experiments conducted by Xiao et al. [2022].
>
> **---The prior work and related concepts upon which the contribution of the authors build is not explained very well. Half of the assumptions and derivations are to be found in other papers, making it almost impossible to fully grasp the paper as a stand alone. It appears to be follow-up work by the authors of previous work, hence the authors don’t seem to find it necessary to fully explain the background concepts**
> The concept of this study is fully explained in Section 4.1. On the other hand, due to page limitations, it is only possible to include some reasoning in the main text. The detailed is provided in Appendix A.
> SAF offers memory reduction by eliminating the need to calculate membrane potentials during training, as well as halving the time required for weight calculation during forward propagation. In our experiments, SAF achieved a significant reduction of 30\% in memory usage compared to OTTT. Similarly, the training time was also reduced by the same percentage, at least. These results demonstrate substantial differences between the two methods and highlight the significance of SAF's contributions. Moreover, we have established the theoretical consistency of SAF with OTTT and spike representation for both of forward and backward. Based on these contributions, our paper goes beyond being a mere follow-up, and we believe that it is suitable for acceptance by ICLR.
>
> **---It is not clear to me why the approach works for small time steps. As stated in section 3.2 Spike representation assumes a rather large latency (T $\to \inf$) to work. Does that not also apply to SAF? In Fig. 5 you show that firing rates for OTTT and SAF are NOT Identical. Then why do your experiments assume T=6? Where does the averaged spike representation come from in practice?**
> There are two reasons why we set $T=6$. First, OTTT's original paper (Xiao et al. [2022]) sets $T=6$. Second, in the case of CIFAR-10, the original paper on spike representation (Meng et al. [2022]) compares the four cases of $T=20,15,10,5$. The results show that the accuracy drop was less than 1\%, and SOTA was achieved at $T=5$. Since SAF-F and OTTT$ _ \rm{A}$ are different at $T=6$, it is unnecessary to increase $T$ to obtain similar results.
> It is indeed consistent with SAF-F for spike representation, but not with OTTT$_ \rm{A}$ . Section 5.2 confirms this distinction, highlighting the differences between the two methods. Consequently, the firing rates are expected to differ, which is correct. On the other hand, Section 5.1 describes the results of an experiment that confirms the equivalence between SAF-E and OTTT$ _ \rm{O}$. The figure presented in this section demonstrates the same firing rates. Thanks to your review, we have added explanations to the figure captions in Section 5, improving readability.
>
> **---In Section 3, Spike Representation, what is x for the weighted average input? Not mentioned before**
> $x$ is the value of the input data. In the case of CIFAR-10, it is the value of each pixel in the image. This explanation has been added to the main text.

---

### Meta-Review · Area_Chair_pbT6 · 2023-12-11

**Metareview:**

This paper presents a new method for reducing memory requirements and speeding up training of spiking neural networks.  Although the ideas seem promising and reviewers appreciated many strengths in the manuscript, they ultimately did not judge it to be above the bar for acceptance to this year's meeting. I'm very sorry that this paper could not be accepted to ICLR this year, and I wish the authors the best of luck in revising this work for publication elsewhere.

**Justification For Why Not Higher Score:**

The reviewers were unanimous that it was below the bar for acceptance.

**Justification For Why Not Lower Score:**

n/a

---

### Decision · Program_Chairs · 2024-01-16

Reject